# Functional analysis across model systems implicates ribosomal proteins in growth and proliferation defects associated with hypoplastic left heart syndrome

Tanja Nielsen[1,2†], Anaïs Kervadec[1†], Jeanne L Theis[3†], Maria A Missinato[1], James Marchant[1], Michaela Romero[1], Katya Marchetti[1], Aashna Lamba[1], Xin-Xin I Zeng[1], Marie Berenguer[1], Stanley M Walls[1], Analyne Schroeder[1], Katja Birker[1], Greg Duester[1], Paul Grossfeld[4], Timothy J Nelson[5], Timothy M Olson[6], Karen Ocorr[1], Rolf Bodmer[1*], Georg Vogler[1*], Alexandre R Colas[1*]

[1]Center for Cardiovascular and Muscular Diseases, Sanford Burnham Prebys Medical Discovery Institute, La Jolla, United States; [2]Department of Biochemistry, Chemistry and Pharmacy, Freie Universität Berlin, Berlin, Germany; [3]Cardiovascular Genetics Research Laboratory, Mayo Clinic, Rochester, United States; [4]University of California San Diego, Rady Children's Hospital, San Diego, United States; [5]Center for Regenerative Medicine, Division of Pediatric Cardiology, Department of Pediatric and Adolescent Medicine, Division of General Internal Medicine, Department of Molecular and Pharmacology and Experimental Therapeutics, Mayo Clinic, Rochester, United States; [6]Department of Cardiovascular Medicine, Division of Pediatric Cardiology, Department of Pediatric & Adolescent Medicine, Cardiovascular Genetics Research Laboratory, Mayo Clinic, Rochester, United States

*For correspondence:
rolf@sbpdiscovery.org (RB);
gvogler@sbpdiscovery.org (GV);
acolas@sbpdiscovery.org (ARC)

†These authors contributed equally to this work

## eLife Assessment

This **important** study applies an innovative multi-model strategy to implicate the ribosomal protein (RP) encoding genes as candidates causing Hypoplastic Left Heart Syndrome. The evidence from the screen in stem cell-derived cardiomyocytes and whole genome sequencing of human patients, followed by functional analyses of RP genes in fly and fish models, is **convincing** and supports the authors' claims. This work and methodology applied would be of broad interest to medical biologists working on congenital heart diseases.

**Abstract** Hypoplastic left heart syndrome (HLHS) is the most lethal congenital heart disease (CHD) whose genetic basis remains elusive, likely due to oligogenic complexity. To identify regulators of cardiomyocyte (CM) proliferation relevant to HLHS, we performed a genome-wide siRNA screen in human iPSC-derived CMs, revealing ribosomal protein (RP) genes as the most prominent effectors of CM proliferation. Whole-genome sequencing of 25 HLHS proband–parent trios similarly showed enrichment of rare RP gene variants, including a damaging RPS15A promoter variant shared in a familial CHD case. Cross-species functional analyses demonstrated that perturbation of RP genes impairs cardiac growth: knockdown of RPS15A, RPS17, RPL26L1, RPL39, or RPS15 reduced CM proliferation, caused cardiac malformations in *Drosophila*, and produced hypoplastic or dysfunctional hearts in zebrafish. Genetic interactions between RP genes and key cardiac transcription factors (TBX5 and NKX2–7) further support their developmental role. Importantly, p53 suppression

or Hippo activation partially rescued RP deficiency phenotypes. Together, these findings implicate RP genes as critical regulators of cardiogenesis and candidate contributors to HLHS.

## Introduction

Hypoplastic left heart syndrome (HLHS) accounts for 2–3% of all cases of congenital heart disease (CHD) *Gordon et al., 2008*; *Reller et al., 2008* and is characterized by underdevelopment of the left ventricle, mitral and aortic valves, and aortic arch (*Crucean et al., 2017*). HLHS has a recognized genetic component based on its familial association with left-sided obstructive CHDs (*Konno et al., 2010*; *Agopian et al., 2017*; *Martin et al., 2018*). However, segregation analyses in multiplex HLHS-CHD families *Martin et al., 2018* and genome-wide association studies of large cohorts *Agopian et al., 2017* have lacked sufficient power to conclusively identify candidate HLHS-susceptibility genes with small to moderate effect sizes. Defects in a small number of genes involved in cardiogenesis, such as *NOTCH1* (*Theis et al., 2015a*), *NKX2–5* (*Elliott et al., 2003*), *MYH6* (*Theis et al., 2015b*), and *GATA4 Elliott et al., 2003*; *Zanon et al., 2017* have been implicated as contributors to HLHS, as well as multiple other CHDs. However, the variety of phenotypic manifestations of HLHS, together with numerous studies linking it to diverse genetic loci (*Theis et al., 2015a*; *Elliott et al., 2003*; *Theis et al., 2015b*; *Dasgupta et al., 2001*), suggests that HLHS is genetically heterogeneous and has a multigenic etiology (*Yagi et al., 2018*; *Yaich et al., 1998*).

It has also been hypothesized that the underdevelopment of the left ventricular (LV) myocardium includes restricted blood flow across the mitral valve and its hemodynamic effect during ventricular growth and development ('no flow – no grow'; *Grossfeld et al., 2009*), as well as endocardial defects (*Miao et al., 2020*). Therefore, HLHS manifestation may be due to a combination of cardiomyocyte (CM) autonomous and non-autonomous genetic as well as mechanical effects, all of which likely affect proliferation and differentiation in the developing heart.

In a digenic HLHS mouse model, loss of both Sin3A Associated Protein 130 (*Sap130*) and the protocadherin *Pcdha9* causes decreased CM proliferation resulting in LV hypoplasia (*Liu et al., 2017*). In addition, LVs from HLHS patients have cardiac damage associated with CM proliferation defects compared to healthy subjects (*Gaber et al., 2013*). These studies suggest that intrinsically defective cardiac differentiation and impaired CM proliferation are likely contributing to HLHS-associated heart defects. Further strengthening this hypothesis, we found that HLHS-patient-derived hPSC-CMs exhibited reduced proliferation compared to the parents (*Theis et al., 2020*). There is a need to identify the potentially causative genes, which contribute to those processes described above and to elucidate their role in HLHS pathogenesis.

To advance CHD gene discovery, we have developed a multi-model systems platform enabling both to rigorously prioritize candidate genes from whole-genome sequencing (WGS) based on rare, predicted-damaging variants and their mode of inheritance, and to functionally characterize gene function in complementary and genetically tractable model systems: human-induced pluripotent stem cells (hPSCs), the fruit fly *Drosophila melanogaster*, and zebrafish *Danio rerio*. The genetic basis of cardiac development, originally uncovered in *Drosophila*, is fundamentally conserved across species (*Bodmer, 1993*; *Bodmer, 1995*; *Cripps and Olson, 2002*; *Bodmer and Frasch, 2010*) and ~80% of human disease genes have fly orthologs (*Bier and Bodmer, 2004*). In addition, the non-redundant fly genome, together with the simple structure and function of the fly heart, allows straightforward genotype–phenotype correlations. Zebrafish (*Evans et al., 2010*; *Bakkers, 2011*; *Liu and Stainier, 2012*), which has a two-chambered heart and can be easily manipulated using morpholino (MO) injections and CRISPR technologies to affect heart development and function. In addition, phenotypes can be directly observed in the developing larva for several days post-fertilization (*Fink et al., 2009*). Finally, CMs differentiated from hPSCs (hPSC-CMs) enable the identification and quantification of cellular phenotypes associated with human diseases, including HLHS (*Theis et al., 2015a*; *Huang et al., 2011*; *Yu et al., 2018*; *Paige et al., 2020*) and are amenable to large-scale functional screens (*Diez-Cuñado et al., 2018*; *Murphy et al., 2021*), thereby enabling them to serve both as a discovery and validation platform. This integrated approach allows for rapidly identifying novel HLHS-associated gene candidates and for characterizing their ability to regulate developmental processes associated with disease (e.g., hypoplasia) (*Theis et al., 2020*).

Defective CM proliferation is likely a hallmark of HLHS (*Liu et al., 2017*; *Theis et al., 2020*; *Gaber et al., 2013*), thus to uncover novel regulators of human CM proliferation with potential links to HLHS, we performed a whole-genome siRNA screen (18,055 genes) using an EdU assay in hPSC-CMs and identified ribosomal proteins (RPs) as the major class of genes controlling CM proliferation. In parallel, WGS of 25 poor-outcome HLHS proband–parent trios followed by unbiased variant filtering revealed an enrichment for rare, predicted-damaging variants in RP genes. Moreover, analysis of a high-value family (75H) comprised of an HLHS proband, his phenotypically normal parents, and a fifth-degree relative born with left-sided CHD led to the identification of a predicted-damaging variant in RP protein gene *RPS15A* that segregated with disease. Consistent with a central role for RPs in the regulation of heart development, heart-specific knockdown (KD) of HLHS-associated RPs in *Drosophila* caused severe phenotypes, including near-complete heart loss and lethality. Similarly, in zebrafish, *rps15a* CRISPR mutants and morphants had greatly reduced heart size and CM number, looping defects, and diminished contractility. In this context, a multi-model system evaluation (hPSC-CMs, flies, and zebrafish) of RP function shows their important role in heart growth and differentiation and CM proliferation by modulating the p53, Myc, and Hippo pathways. Moreover, we find evidence that RPs may regulate heart development and CM proliferation in a cardiac-specific manner by synergistically interacting with core cardiac transcription factors (cTFs) such as T-box, Nkx, and Gata factors. In sum, here we show that RP genes are potent regulators of CM proliferation and cardiac growth and differentiation, consistent with a potentially critical role in the hypoplastic phenotype of HLHS. We suggest that RP genes are an emerging class of novel genetic effectors in HLHS.

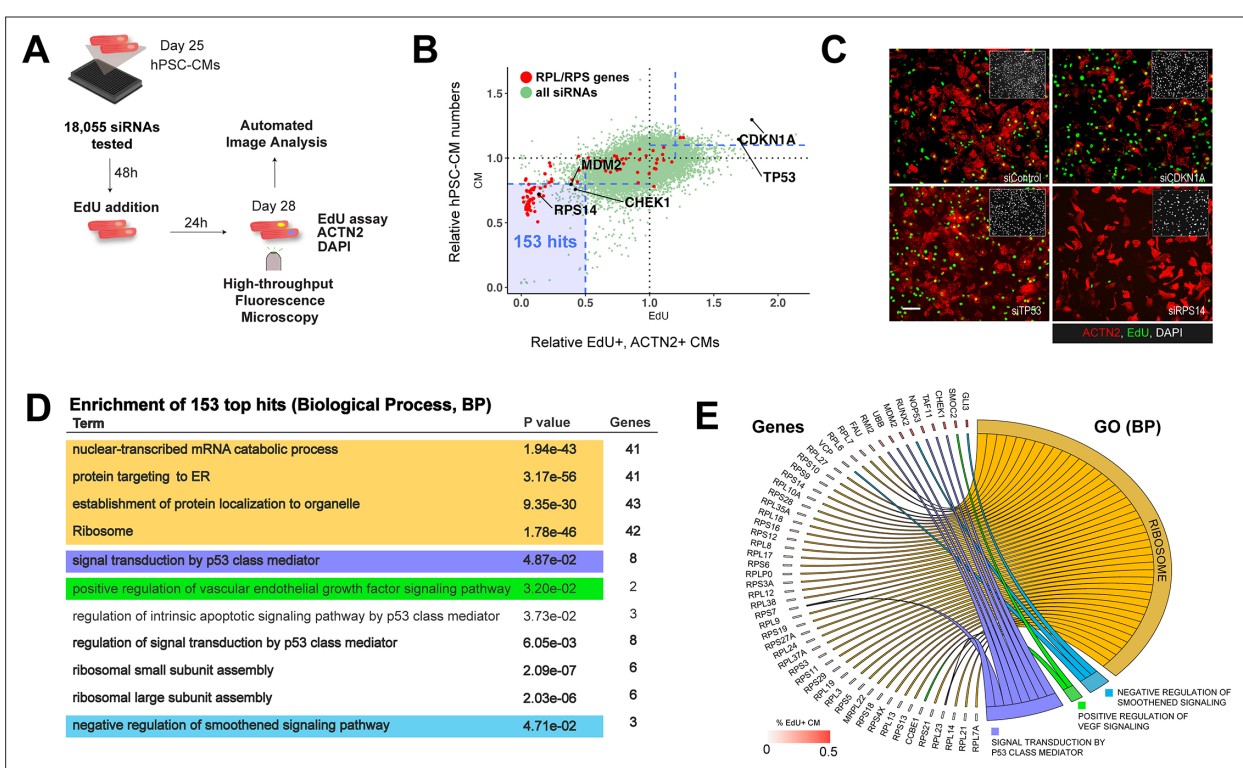

**Figure 1.** Whole-genome siRNA screen identified ribosomal proteins as agonists of cardiomyocyte (CM) proliferation. (**A**) High-throughput iPSC-derived CM proliferation screen overview. (**B**) Screen result showing normalized % EdU+ CMs (X-axis) and relative total number of CMs (Y-axis) upon knockdown of genome-wide siRNAs (18,055 siRNAs). siRNAs for RPL and RPS genes highlighted in red. (**C**) Representative immunofluorescence images of proliferation (EdU incorporation, green, CM marker ACTN2, red) of induced hPSC-CMs upon *TP53* and *RPS14* knockdown. Insets: nuclei (DAPI). (**D**) Gene ontology enrichment analysis for whole-genome sequencing (WGS) hits (BP, biological process; FDR-corrected analysis using gprofiler2). (**E**) Overview of hits corresponding to top 4 non-redundant BP categories.

The online version of this article includes the following figure supplement(s) for figure 1:

**Figure supplement 1.** Functional validation of RP–mediated control of proliferation in hiPSC-CMs and human dermal fibroblasts.

# Results

## Whole-genome siRNA screen identifies RPs as central regulators of CM proliferation

Based on previous and our recent data (*Liu et al., 2017*; *Gaber et al., 2013*; *Theis et al., 2020*), impaired CM proliferation is emerging as an important mechanism in HLHS pathogenesis. Thus, to comprehensively map the human genome for novel regulators of CM proliferation, we screened a library of 18,055 siRNAs for their ability to modulate proliferation in human iPSC-derived CMs (hPSC-CMs). To assess CM proliferation, we used a dual read-out and quantified both DNA synthesis using EdU incorporation (*Diez-Cuñado et al., 2018*) and total number of CMs, 3 days after treatment (*Figure 1A–C*). Briefly, siRNAs were transfected into 25-day-old hPSC-CMs (*Yu et al., 2018*; *Cunningham et al., 2017*). After 2 days, EdU was added to the wells for 24 hr. On day 3, cells were fixed and co-stained for sarcomeric protein alpha-Actinin (ACTN2), EdU, and DAPI. Next, the number of EdU/alpha-Actinin double-positive cells and the total number of CMs were determined using a commercially available high-throughput image analysis software (MetaXpress, Molecular Devices) (*Diez-Cuñado et al., 2018*). Note that alpha-Actinin-, EdU+ cells which typically express fibroblast markers (i.e POSTN) (*Cunningham et al., 2017*), were excluded from this analysis, to focus on CM biology only. In total, we found 153 siRNAs that decreased proliferation (<0.5-fold EdU incorporation) and number of CMs (<0.8-fold), and 162 siRNAs that increased EdU (>1.2-fold) and CM number (>1.1-fold; *Figure 1B, C* and *Supplementary file 1*). Consistent with previous studies (*Zhang et al., 2003*; *Yuan et al., 2014*), KD of cell cycle checkpoint agonist, CHEK1 and MDM2, the negative regulator of TP53, both were among top hits that reduced CM proliferation. As expected, we identified TP53 and CDKN1A among the genes that significantly increased CM cell number upon KD (*Figure 1B, C*). Next, to gain insight into the molecular pathways regulating proliferation in hPSC-CMs, we performed gene ontology (GO) term analysis of hits decreasing EdU incorporation or CM numbers and found an enrichment for genes associated with 'translation/ribosome' and p53 signaling as top hits (*Figure 1D, E*). Consistent with these observations, we had previously found that the p53 pathway is dysregulated in hPSCs differentiated from an HLHS family trio (*Theis et al., 2020*). Remarkably, the most represented gene family is RPs, KD of which caused consistently the strongest inhibition of cell proliferation (*Figure 1B*). Next, since the primary screen was performed as a single data point, we next sought to validate these findings by re-testing all 80 RP genes for function (*Zhou et al., 2015*) using a distinct set of siRNAs in biological quadruplicate conditions. Remarkably and consistent with the primary screen, KD of 59/80 RPs significantly reduced proliferation as compared to siControl (p < 0.05; *Figure 1— figure supplement 1A, B*). In this context, further functional testing in fibroblasts revealed that the antiproliferative effect of RP loss of function is cell type independent (*Figure 1—figure supplement 1C, D*). Together, these findings establish RP function as essential for cell proliferation, including in hPSC-CMs, and suggest that RPs may represent a previously unrecognized class of regulators of ventricular growth during embryonic development.

## Enrichment of RP gene variants in an HLHS patient cohort with poor clinical outcome

To identify candidate genes involved in HLHS pathogenesis, we performed WGS, variant filtering, and enrichment analysis (see also, *Theis et al., 2020*), and Methods: WGS and bioinformatic strategies in a cohort of 25 HLHS patient–parent trios with poor clinical outcome defined as either prenatal – restrictive atrial septal defect (*n* = 2); postnatal – reduced right ventricular ejection fraction following stage II or III surgical palliation (*n* = 19); protein-losing enteropathy (*n* = 2); or cardiac transplantation/ failed surgical palliation (*n* = 2). The 25 probands were comprised of 18 males and 7 females, 24 of whom were of white ancestry. The HLHS phenotype with respect to valve morphology (mitral atresia, MA; mitral stenosis, MS; aortic atresia, AA; aortic stenosis, AS) was MA/AA in 10, MS/AS in 9, MA/ AA in 4, MA/AS in 1, and unknown in 1. CHD in the parents was excluded by echocardiography in all but two families, in which a bicuspid aortic valve was present in the mother. Genomic sequences were filtered for uncommon (minor allele frequency [MAF] <1%) de novo and recessive variants in coding and regulatory regions of genes expressed in the developing heart and predicted to alter protein structure or expression, yielding 292 unique HLHS candidate genes that primarily fit a recessive mode of inheritance (*Figure 2A* and *Supplementary file 2*). To determine whether certain gene networks

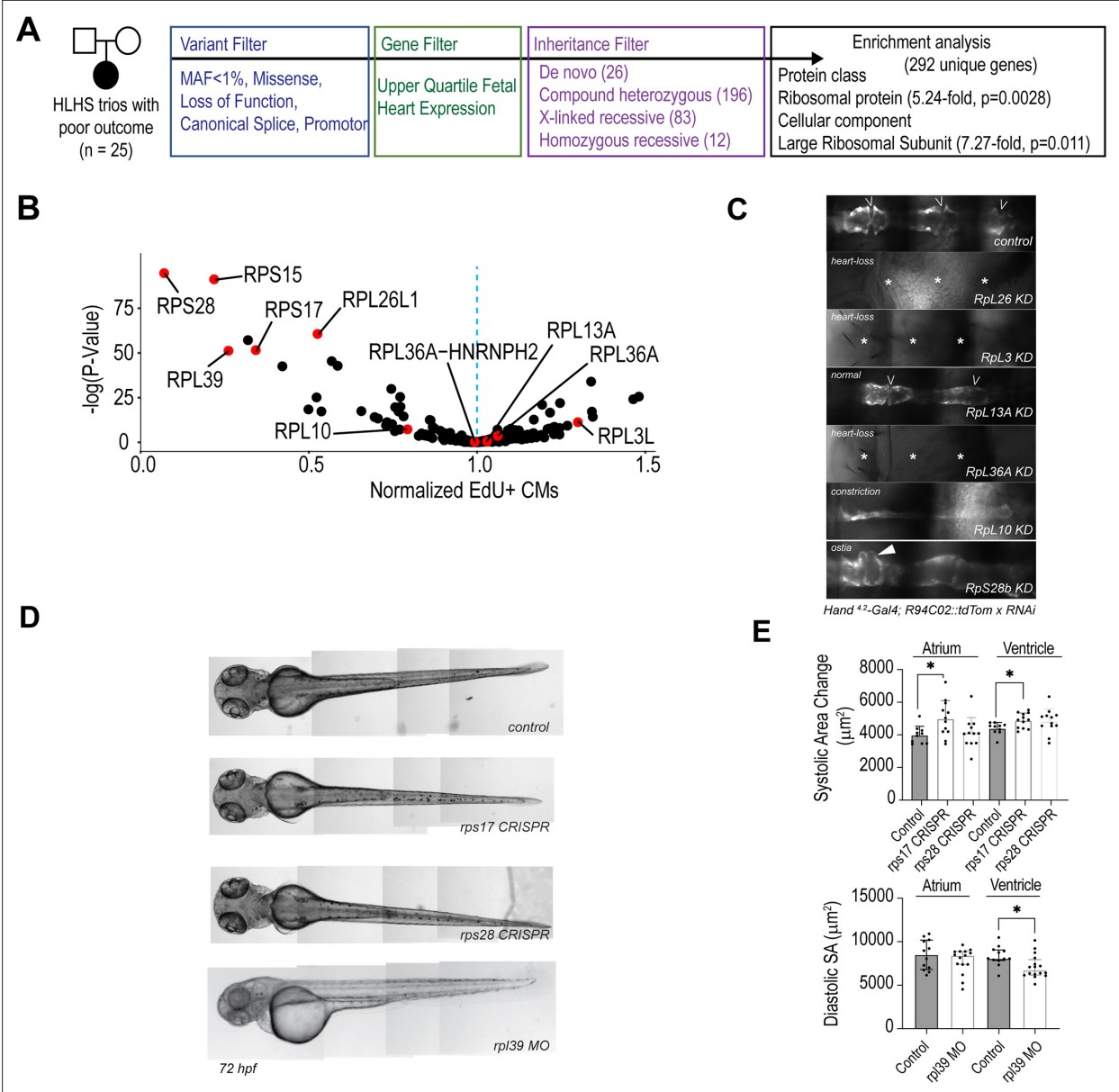

**Figure 2.** Ribosomal gene variants identified in hypoplastic left heart syndrome (HLHS). (**A**) Gene prioritization scheme of 25 poor-outcome proband–parent trios. (**B**) Testing 292 HLHS candidate genes from all poor-outcome families in cardiomyocytes (CMs) identified RPs as major regulators of hPSC-CM proliferation (normalized fraction of ACTN2+/EdU+ cells). (**C**) *Drosophila* cardiac phenotypes induced by loss of RP genes affected in HLHS patients with poor outcome. The heart is visualized by RFP expression specifically in CMs (R94C02::tdTom). Knockdown is achieved by sustained Gal4/UAS activity using Hand4.2-Gal4. (**D**) Wild-type zebrafish larva, *rps17* and *rps28* CRISPR mutants, and *rpl39* morphants at 72 hpf. *rpl39* morphants, injected with 1 ng MO in lateral view, show mild edema. (**E**) Systolic surface area (SA) upon *rps17* and *rps28* CRISPR, and diastolic SA after *rpl39* MO in the atrium and ventricle of zebrafish hearts.

were over-represented among these variants, we used two online bioinformatics tools (STRING and PANTHER; *Mi et al., 2019*; *Szklarczyk et al., 2019*). After applying Fisher's exact test and false discovery rate corrections, RPs were the most enriched class of proteins when compared to the reference proteome, which includes data annotated by protein class (5.24-fold, p = 0.0028), and cellular component (7.27-fold, p = 0.011). In total, 14 variants found in 9 RP genes among 6 HLHS probands were identified, most fitting a recessive inheritance disease model (*Table 1*). These RP variants encompass mutations in upstream promoter regions that potentially affect transcription factor-binding sites, as well as non-synonymous substitutions inside the protein coding regions.

**Table 1.** Variants in ribosomal genes in hypoplastic left heart syndrome (HLHS) probands.

| Gene | Proband (age, sex) | Mode of inheritance | Variant | Type | MAF% | CADD score | TFBS affected | hPSC-CM proliferation | Fly gene and defects | Zebrafish gene and defects | Patient outcome |
|---|---|---|---|---|---|---|---|---|---|---|---|
| RPL26L1 | 145H (3y, m) | Compound heterozygous | −1248A>G; V97M | Regulatory; missense | 0.029; 0.032 | −; 21.3 | Pdx1; NFE2L1::MAFG; FOXC1 | Reduced | RpL26: lethal, no heart | rpl26: n.t. | Restrictive ASD, PLE |
| RPL36A | | X-linked recessive | −1321C>T | Regulatory | 0.055 | - | PAX2 | No effect | RpL36A: lethal, no heart | rpl36a: n.t. | |
| RPS15 | | Compound heterozygous | −1558C>T; T101S | Regulatory; missense | 0; 0.102 | −; 23.5 | FOXC1 | Reduced | RpS15: lethal | rps15: n.t. | |
| RPL39 | 151H (20y, m) | X-linked recessive | −1359T>C | Regulatory | 0.653 | - | HOXA5 | Reduced | RpL39: lethal | rpl39: morphants mild edema, reduced ventricular size | Reduced RV function |
| RPL3L | 96H (22m, m) | Compound heterozygous | R200Q; R242W | Missense; missense | 0.966; 0.432 | 21.2; 14.94 | - | Elevated | RpL3: no heart | rpl3: n.t. | Reduced RV function |
| RPL13A | 201H | Compound heterozygous | −92–645C>T; −29–191C>T | Regulatory; regulatory | 0.72; 0.046 | - | FOXD1; GATA2; ETS1; ELF5; FOXC1 | No effect | RpL13A: no phenotype | rpl13a: n.t. | Failing Fontan circulation, transplant at 14y |
| RPS17 | 325H | Homozygous recessive | S136N | Missense | 0 | <10 | - | Reduced | RpS17: lethal | rps17: CRISPR mutants show systolic atrial dysfunction, shortened heart period | Reduced RVEF and increased RVEDP at 9m |
| RPL10 | 76H | X-linked recessive | 24–218G>A | Regulatory | 0.583 | - | ELK1; ETS1; SPIB; POLR2A; HEY1; Hltf | Reduced | RpL10: constricted | rpl10: n.t. | Reduced RVEF at 9y |
| RPS28 | | Compound heterozygous | −589G>A; −505A>T | Regulatory; regulatory | 0.061; 0.08 | - | CTCF | Reduced | RpS28b: ostia defect | rps28: CRISPR mutants show no heart phenotype | |

MAF – minor allele frequency; TFBS – transcription factor-binding site; n.t. – not tested; ASD – atrial septal defect; PLE – protein-losing enteropathy; RV – right ventricle; RVEF – right ventricular ejection fraction; RVEDP – right ventricular end diastolic pressure.

Next, all 292 prioritized candidate genes were systematically evaluated for effects on the proliferation of hPSC-CMs in vitro and for cardiac differentiation of fly hearts in vivo. Again, loss of RP function (*RPL39*, *RPL26L1*, *RPS15*, *RPS17*, *RSPS28*) caused the most severe phenotypes on proliferation in hPSC-CMs (*Figure 2B*, *Supplementary file 3*), which was consistent with our genome-wide proliferation screen (see *Figure 1B*). In the fly, KD of RPs in the cardiac lineage and throughout embryonic development was achieved using a *Hand*[4.2]-Gal4 driver (see *Brand and Perrimon, 1993* and methods) and led to partial or complete heart loss (*RpL26*, *RpL36A*, *RpL3*) or caused lethality (*RpS17*, *RpL39*, *RpS15*, *RpL26*, *RpL36A*) (*Figure 2C*). Also, KD of *RpL10* gave rise to severely constricted hearts, while KD of *RpS28b* caused inflow tract defects (*Figure 2C*). Finally, functional testing of RPs in zebrafish embryos revealed that KD of *rpl39* leads to reduced ventricle size, whereas *rps17* CRISPR mutants exhibit systolic dysfunction (*Figure 2D, E*). In sum, our patient-centric approach (cohort of 25 probands) identifies RPs as the most enriched gene category affected by HLHS-associated variants (STRING, PANTHER analysis). In this context, systematic testing in multi-model systems (fly, zebrafish, hPSC-CMs) identifies RPs as most potent regulators of cardiac differentiation, proliferation, and contractility among HLHS-associated genes. Collectively, these findings highlight that RPs regulate critical steps of cardiogenesis that are also found to be defective in HLHS probands (*Gaber et al., 2013*; *Liu et al., 2017*) and thus suggest a potential link between RP biology and HLHS phenotypic etiology.

## RPS15A variant associated with HLHS in familial CHD

To further establish a phenotypic link between RP function and HLHS-associated phenotypes, we selected a rare familial CHD case (75H) (*Figure 3—figure supplement 1A*), that is comprised of a young teenager with HLHS (MS/AS) and normal ventricular function following Fontan operation, his phenotypically normal parents, and a fifth-degree female relative born with left-sided CHD (bicuspid aortic valve and coarctation of the aorta). Based on these observations, we hypothesized the presence of a heterozygous driver variant exhibiting incomplete penetrance and variable expression. To investigate further, we filtered WGS data for rare, predicted-damaging variants in coding and regulatory regions. This analysis identified six prioritized variants (*Supplementary file 4*), including one located in the *RPS15A* gene locus, shared by the HLHS proband, the unaffected mother, and the proband's fifth-degree relative with CHD.

We next investigated whether family members carrying these variants (the mother and proband) exhibit defects in CM proliferation compared to the father. Notably, analysis of EdU incorporation in CMs from the parent–proband trio revealed significantly reduced proliferation in both the proband and mother compared to the father. However, the proband displayed a markedly more severe phenotype than the mother (*Figure 3A, B*). Interestingly, this reduction in proliferation appeared to be specific to CMs, as EdU incorporation in the non-CM cell population showed no significant differences among the family members (*Figure 3—figure supplement 1B*). Collectively, these findings suggest a phenotypic association between the presence of the variants and impaired CM proliferation.

Finally, to determine the potential relative contribution of the six genes harboring rare and damaging variants in the regulation of CM proliferation, we performed functional KD experiments in hPSC-CMs in vitro. Remarkably, while KD of 5 of the other prioritized genes from the 75H family also resulted in moderate reduced CM proliferation, RPS15A KD produced the most pronounced effect, with over an 85% reduction in proliferation, emerging as the top candidate (*Figure 3C, D*). Together, these findings indicate that RPS15A plays a critical role in regulating CM proliferation in human pluripotent stem cells (hPSCs) and may influence heart developmental processes that are disrupted in HLHS.

## RPS15A KD in *Drosophila* and zebrafish causes severe cardiac proliferation and differentiation defects

To characterize the role of *RPS15A* during heart development, we first induced a heart-specific KD of *RpS15Aa* in flies (see methods), which caused a partial or complete loss of the heart (*Figure 3E, F*). Note that this phenotype is consistent with our results above and a previous study involving another RP gene (*RPL13*) (*Schroeder et al., 2019*).

Next, to evaluate the role of *RP* genes during heart development, we knocked out *rps15a* in zebrafish (*D. rerio*) using CRISPR (*Evans et al., 2010*; *Bakkers, 2011*; *Liu and Stainier, 2012*). We examined F$_0$ larva at 72 hpf (hours post fertilization) using high-speed digital imaging (SOHA; *Fink*

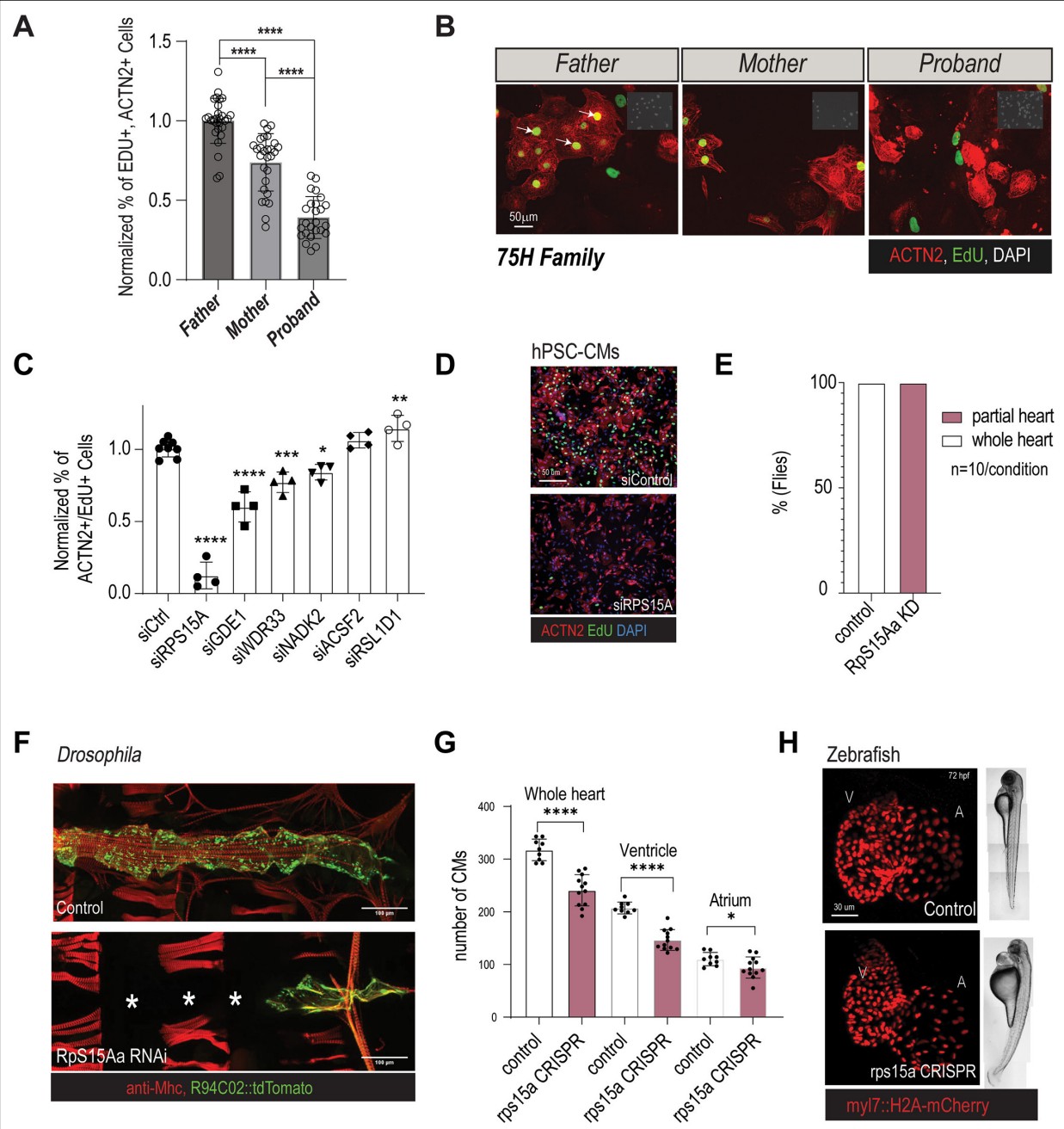

**Figure 3.** Characterization of RPS15A from the 75H hypoplastic left heart syndrome (HLHS) index family. (**A–C**) Prioritized candidate genes from the 75H family and relative hPSC-cardiomyocyte (CM) proliferation capacity upon KD. (**D**) Representative immunofluorescence images of proliferation (EdU incorporation, green; CM marker ACTN2, red) of induced hPSC-CMs upon siRPS15A knockdown. (**E**) Heart-specific KD of *RpS15Aa* in *Drosophila* adult hearts causes loss of heart tissue and is fully penetrant. (**F**) Representative immunofluorescence images of control and *RpS15Aa*-RNAi show partial heart loss (myosin heavy chain, Mhc, red; heart tissue-reporter, green). (**G**) CRISPR-mediated loss of *rps15a* in $F_0$ larval zebrafish hearts causes decrease in CM number. (**H**) Representative immunofluorescence of hearts and whole-mount images of control and *rps15a*-CRISPR $F_0$ larval hearts (CM nuclei reporter, red).

The online version of this article includes the following figure supplement(s) for figure 3:

**Figure supplement 1.** Model organism phenotypes caused by loss of RP genes.

**Figure supplement 2.** *rps15a* zebrafish larval morphants show heart dysfunction and reduction in cardiomyocyte (CM) number.

**Figure supplement 3.** Proliferation and heart function depend on RPS15A and are regulated via TP53 in cardiomyocytes (CMs) and zebrafish.

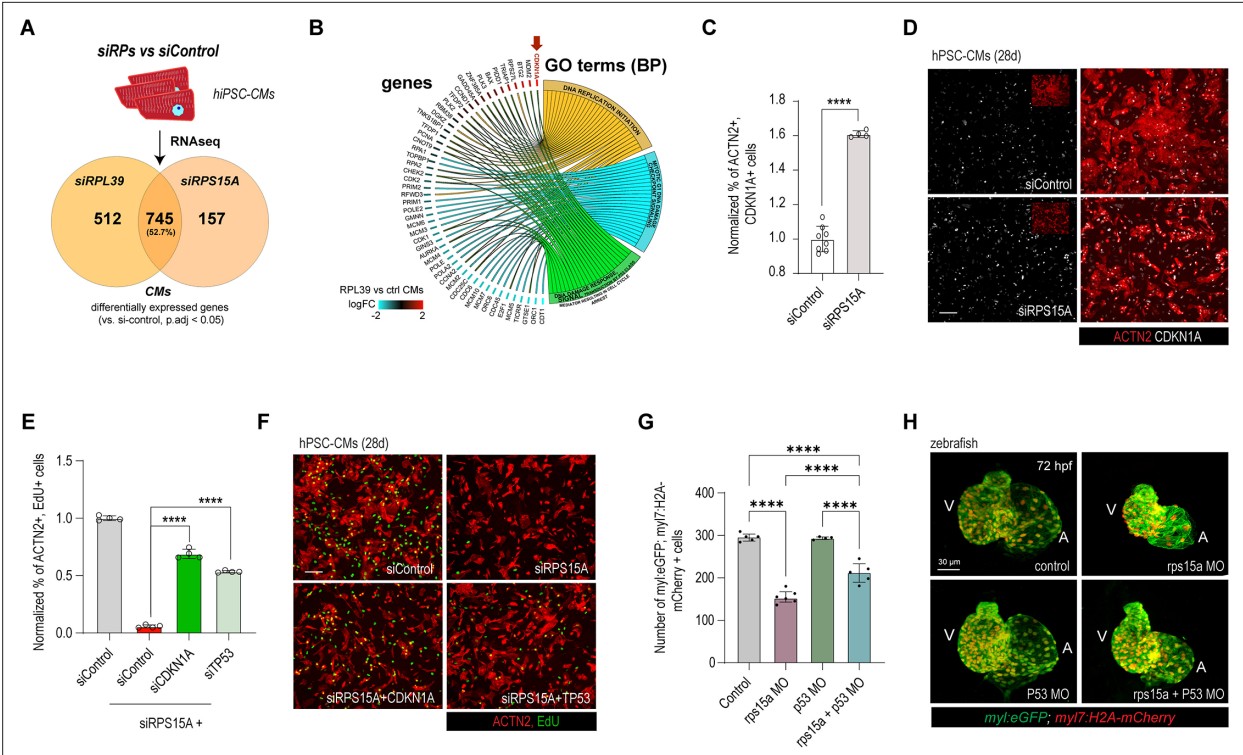

**Figure 4.** Loss of ribosomal gene function in cardiomyocytes (CMs) invokes TP53-stress response. (**A**) RNA-sequencing of hPSC-CMs following siRNA treatment for *RPL39* and *RPS15A* shows both convergent and divergent transcriptomic response. (**B**) Gene ontology (GO) term analysis of differentially expressed genes following RP KD shows TP53-mediated response, including upregulation of *CDKN1A*. (**C, D**) CDKN1A is highly upregulated in CMs following *siRPS15A* treatment. (**E, F**) Reduced CM proliferation upon *RPS15A* KD is mediated by *CDKN1A/TP53* and can be rescued upon their co-KD. (**G, H**) Larval zebrafish CM number is reduced by morpholino treatment for rps15a and can be attenuated by P53 morpholino co-injection. Control and morphant (MO) hearts of 72 hpf zebrafish larva stained with *Tg(myl7:EGFP)* and *Tg(myl7:H2A-mCherry)*. Note that the smaller heart with aberrant looping by *rps15a* MO is partially reversed by *p53* co-KD. Student's *t*-test, *p < 0.05, **p < 0.01, ***p < 0.001, ****p < 0.0001.

*et al., 2009*, see also methods) and confocal microscopy to monitor heart size and CM numbers, respectively. Remarkably, *rps15a* mutant hearts were smaller in size (*Figure 3—figure supplement 1C, D*), with reduced CM numbers (*Figure 3G, H*). We observed similar effects in response to injection of a *rps15a* MO compared to uninjected larva (*Uechi et al., 2006*; *Ikeda et al., 2017*; *Figure 3—figure supplement 2A–D*). In this context, we also noted that the overall body plan and morphology of *rps15a* CRISPR mutants or morphants were largely unaffected, except for a mild pericardial edema, a possible indicator of heart dysfunction (*Figure 3H*; *Miura and Yelon, 2011*).

We quantified CM number using nuclear markers in Tg(*myl7:eGFP*); Tg(*myl7:H2A-mCherry*) embryos 72 hpf. CM numbers were predominantly reduced in the ventricle compared to the atrium (*Figure 3G, H*). We also assessed total cardiac cell numbers and cardiac cell proliferation by phospho-histone H3 (PH3) immunostaining and DAPI to quantify total cardiac cells. We again found that the total cell numbers were significantly reduced in both rps15a morphants and crispants (*Figure 3—figure supplement 3A–C*). Importantly, the proportion of proliferating (PH3+) cardiac cells was also decreased in rps15a morphants and crispants at both 24 and 48 hpf, but not at 72 hpf when proliferation normally declines (*Figure 3—figure supplement 3A–C*). Collectively, these data support the hypothesis that *RPS15A* is required for CM proliferation and heart morphogenesis (hPSC-CMs, fly, and zebrafish), thus suggesting a potential role for RPs as genetic drivers of hypoplasticity observed in HLHS.

## RPs control CM cell cycle by regulating TP53 pathway activity

To investigate how RPs regulate cell cycle activity in hPSC-CMs, we performed RNA-seq upon KD of *RPS15A* and *RPL39*, which belong to the small and the large ribosomal subunits that harbor predicted-damaging variants in 75H and 151H probands, respectively. Comparison of differential

gene expression between *siRPS15A* and *siRPL39* KD revealed that ~53% of genes were commonly dysregulated (*Figure 4A*), indicating the existence of a conserved RP-dependent transcriptional network. Consistent with a central role for RPs in the regulation of proliferation, GO term analysis for differentially expressed genes revealed an enrichment for genes involved in DNA replication, mitosis, and DNA damage response mediated by p53, and included the downregulation of positive regulators of cell cycle such as *CDK1*, *CCNA2*, *AURKA*, *PCNA*, and the upregulation of cell cycle arrest mediators such as *CDKN1A*, *BTG2*, *GADD45A* (*Figure 4B* and *Supplementary file 5 and 6*). To confirm these findings, we performed immunostaining for canonical p53 downstream transcriptional target (*Fischer, 2017*), *CDKN1A*, and observed that the percentage of CDKN1A+ CMs was increased by ~60% in response to *RPS15A* KD (*Figure 4C, D*), indicating that RP KD induces cell cycle arrest *via* the activation of the p53 signaling pathway in hPSC-CMs.

Next, to evaluate if the reduction in proliferation caused by RP KD is mediated by the p53 signaling pathway in CMs, we co-KD *TP53* or *CDKN1A* along with *RPS15A* and observed a significant rescue of CM proliferation by EdU incorporation and CM number in comparison to RPS15A KD (*Figure 4E, F* and *Figure 3—figure supplement 3D, E*).

Next, we tested if RP-mediated defects in zebrafish can also be rescued by inhibition of p53. To this aim, we used MOs to co-KD *p53* with *rps15a* (*Robu et al., 2007*) and examined larval hearts at 72 hpf. *p53* MO-mediated KD on its own had little effect on heart function but partially reversed the *rps15a* MO-mediated reduction in heart size and CM numbers (*Figure 4G, H*). Moreover, bradycardia, heart looping, and reduced contractility phenotypes of *rps15a* morphants were also rescued by co-KD of *p53* (*Figure 3—figure supplement 3F, G*). Collectively, these observations support our hypothesis that RPs control cardiac development, including heart size and function, via a p53 pathway-mediated regulation of CM proliferation.

## Heart loss in *Drosophila* by RpS15Aa KD is partially rescued by Hippo pathway activation

The fly heart develops similarly to the vertebrate heart at early stages (*Bodmer, 1995*) and a link between RPs and cell cycle-regulating pathways, such as p53 and Hippo pathways (*Furth et al., 2018*), has been proposed to be conserved between mammals and flies (*Baker et al., 2019*). To test whether p53 has a function in fly heart development similar to hPSC-CMs and zebrafish, we performed cardiac co-KD of *RpS15Aa* and *p53* and examined heart structure and function. Co-KD did not rescue the cardiac defects of *RpS15Aa* KD, suggesting that *p53* has a different role in flies, as compared to vertebrates, as has been reported previously (*Ollmann et al., 2000*).

In vertebrates, TP53 is known to act through the negative regulation of the Hippo pathway (*Raj and Bam, 2019*). In this context, YAP, a downstream effector of Hippo, has been shown to promote CM proliferation in the mouse heart (*Monroe et al., 2019*). We therefore tested if YAP/yorkie can substitute the function of p53 in the fly heart and found that overexpressing the Hippo pathway gene *yorkie* (*yki*), the fly ortholog of YAP, in the developing heart considerably restored *RpS15Aa* KD-induced heart loss in *Drosophila* (*Figure 5A', B*). We also observe the formation of adult ostial structures, which indicates that the larval heart underwent partial remodeling to an adult heart during metamorphosis. Further, we found that *yki*-mediated rescue depends on the transcriptional co-factor encoded by *scalloped* (*sd*, *Drosophila* TEAD1/2/3/4), since upon *sd* KD, *yki* OE can no longer rescue the *RpS15Aa* KD-induced heart loss (*Figure 5A'', B*). Moreover, KD of *sd* in an *RpS15Aa* KD background (without *yki* OE) worsens the *RpS15Aa* KD phenotype (*Figure 5A'', B*).

## *RPS15A* genetically interacts with cTFs *nkx2.7/tinman*, and TBX5/*tbx5a/Dorsocross*, *nkx2.7/tinman*, and *Gata4,5,6/pannier* in model systems

The profound importance of RPs for cell growth and proliferation in most cell types (*Kang et al., 2021* and *Figure 1—figure supplement 1*) raises the question of how tissue-specific phenotypes such as those observed in HLHS can arise from the defective function of ubiquitously expressed genes. To address this apparent paradox, we hypothesized that RPs might functionally interact with tissue-specific proteins such as cTFs to control CM proliferation. Analysis of the impact of cTFs alone on hPSC-CM proliferation from our whole-genome screen revealed that KD of most cTFs did not cause major proliferation defects (except for, e.g., *HES4* and *HOPX*; *Figure 6A*). Among the cTFs, TBX5 is

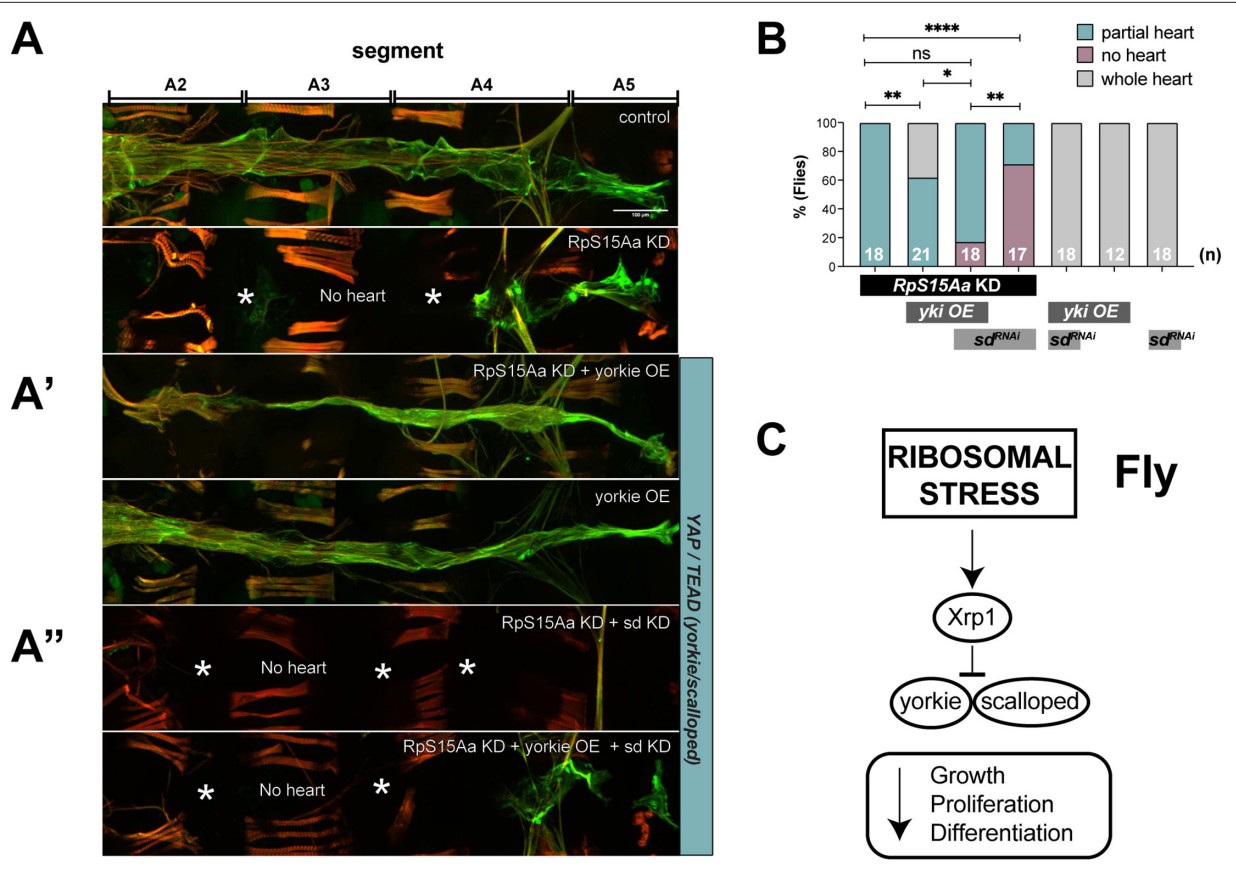

**Figure 5.** Rescue of *RpS15Aa* KD-mediated heart tube loss in *Drosophila* by YAP/yorkie overexpression depending on its co-factor TEAD/scalloped. (**A**) Representative images of RFP-expressing fly hearts. RpS15Aa KD-mediated heart tube loss can be partially rescued by overexpression of *yorkie* (*RpS15Aa* RNAi + *yorkie* OE). The rescue by *yki* OE depends on its co-factor *sd*. Flies were raised at 25°C. (**B**) Quantification of events presented as a percentage of flies exhibiting whole heart tube versus partial heart loss (defined as 25–75% heart tube length compared to wildtype) or no heart tube. Statistics: Fisher's exact test, *p < 0.05. (**C**) Proposed signaling cascade underlying cardiac growth, proliferation, and differentiation impairment following ribosomal stress (adapted from ***Baker et al., 2019***).

specifically expressed in the left ventricle at stages of intense CM proliferation (***Bruneau et al., 1999***) and thus we next asked if *TBX5* could functionally interact with RPs. Remarkably, while hPSC-CM proliferation was not affected by increasing doses of siTBX5 (0–2 nM), increasing si*TBX5* dosage in siRP backgrounds led to proportional decrease in EdU incorporation (two-way ANOVA; ***Figure 6B*** and ***Figure 6—figure supplement 1A, B***), thereby indicating that *RPs* and *TBX5* genetically interact to regulate proliferation in hPSC-CMs.

Conservation of genetic interactions between human and fly cTFs and constitutive genes has been identified before (***Qian et al., 2011***). We therefore specifically tested for genetic interaction between *RpS15Aa* and the cardiac TFs *tinman/NKX2–5*, *pannier/GATA4/5/6*, and *Dorsocross/TBX5* in adult *Drosophila* hearts. A heterozygous deficiency covering *RpS15Aa* causes moderate dilation of hearts compared to controls (***Figure 6—figure supplement 1C***). However, when *RpS15Aa* was placed in trans to loss of function alleles of cardiac TFs *tinman*, *pannier*, or *Dorsocross1/2/3*, these hearts exhibited significantly deformed hearts, which is only rarely observed in the single heterozygotes (***Figure 5C, D***). In addition, *RpS15Aa* heterozygous flies showed a prolonged heart period when combined with a *tinman* or *pannier* heterozygous mutations (***Figure 6—figure supplement 1C***).

Next, we tested whether similar genetic interactions are conserved in zebrafish. Zebrafish express the NKX2–5 ortholog *nkx2.7* in the heart, and functional studies have shown that both *nkx2.5* and *nkx2.7* play critical roles in cardiac development (***Lee et al., 1996***). To determine whether *rps15a* and *nkx2.7* or *tbx5a* genetically interact in zebrafish, we injected low doses of *rps15a* (***Uechi et al., 2006***) MO, in combination with *nkx2.7* (***Targoff et al., 2008***) or *tbx5a* MO. *nkx2.7* MO alone had little

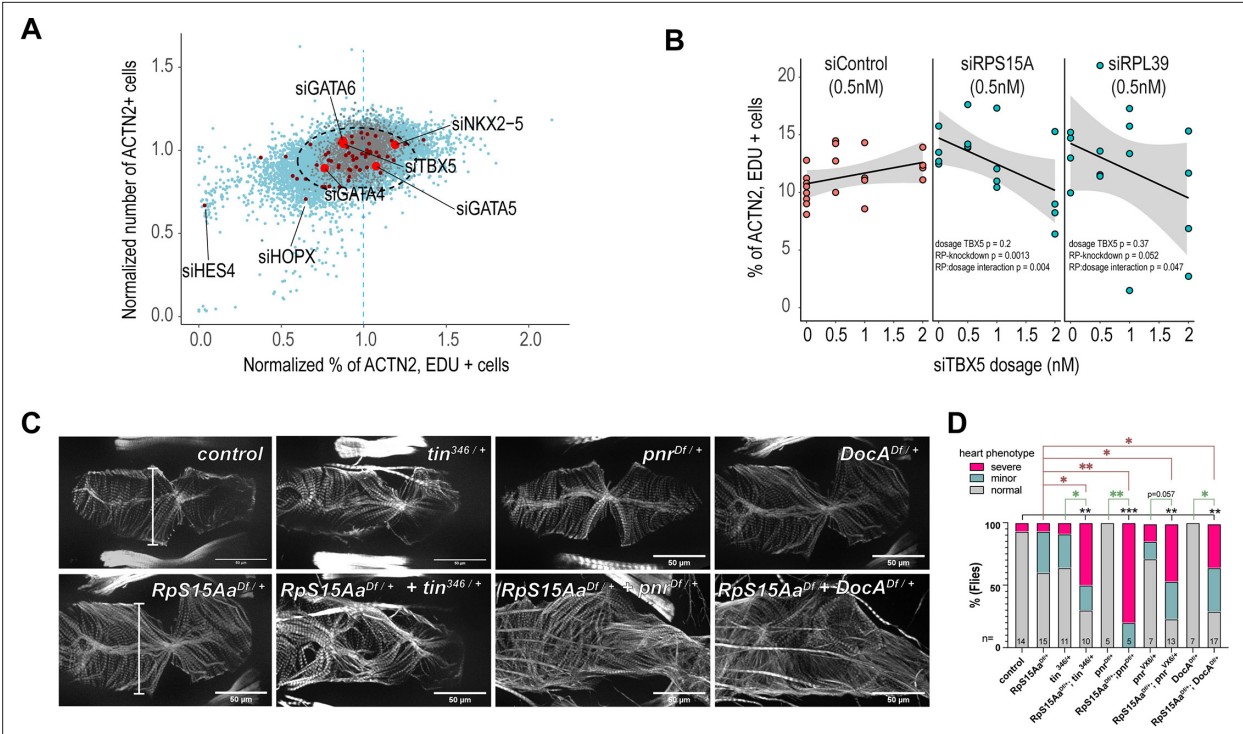

**Figure 6.** *RPS15A genetically interacts with cardiac transcription factors. (**A**) The majority of cardiac transcription factors do not impact cardiomyocyte (CM) proliferation, except for, e.g., HES4 and HOPX. (**B**) *TBX5 genetically interacts with *RPS15A* and *RPL39*. *siTBX5* does not impact CM proliferation at 0.5, 1, 1.5, or 2 nM si-concentration, and neither do *siRPS15A* and *siRPL39* alone at 0.5 nM. CM proliferation is reduced in siRP backgrounds with increased titration of *siTBX5*. Two-way ANOVA for si*TBX5* dosage, RP-knockdown, and their interaction. (**C**) Representative fly heart segment (A4) from control flies, heterozygous mutants (tin[346/+], pnr[VX6/+], Doc[Df/-], Df(RpS15Aa)[+/-]) and transheterozygous mutants. tin[+/-] = tin[346]/+, pnr[+/-] = Df(pnr)/+, Doc[+/-] = Df(DocA)/+. Note the deformation and myofibrillar disorganization in the transheterozygous mutants. (**D**) Quantification of adult *Drosophila* heart defects and genetic interaction. Statistics: Fisher's exact test on absolute numbers testing *normal* versus *severely deformed* hearts. *p < 0.05, **p < 0.005, ***p < 0.001.

The online version of this article includes the following figure supplement(s) for figure 6:

**Figure supplement 1.** Genetic interaction between *RpS15Aa/RPS15A* and *cabeza/EWSR1*.

---

effect on heart size, cross-sectional area, or contractility, whereas double morphants exhibited cardiac dysfunction, with contraction virtually abolished in some animals (***Figure 6—figure supplement 1D***). Furthermore, we observed a significant prolongation of the heart period in double morphants (*rps15a + nkx2.7* or *rps15a + tbx5*a) in comparison to each MO alone (two-way ANOVA; ***Figure 6—figure supplement 1D, E***) consistent with the effects observed in *Drosophila*. Together, these results highlight the existence of evolutionarily conserved (fly, zebrafish, and hPSC-CMs) genetic interactions between RPs and cardiac TF genes in the regulation of cardiogenesis. These data also illustrate how the activity of ubiquitously expressed genes such as RPs can be modulated by cell type-specific genes, like cardiac TFs, to achieve tissue-specific outcomes.

## RP-regulated HLHS-associated genes control proliferation in a CM-specific manner

Previous studies and this work show that reduced RP function can lead to tissue-specific developmental defects, that in turn can be rescued by loss of p53 (***Tiu et al., 2021***; ***McGowan et al., 2011***). While these findings highlight a downstream role for p53 in the development of RP-dependent phenotypes, they also imply that RPs can regulate cell cycle and/or p53 activity in a tissue-specific manner. Thus, to delineate how RPs might control proliferation in a CM-specific manner, we compared differential gene expression profiles upon the two RP KDs in hPSC-CMs and undifferentiated hPSCs (***Figure 7A***). Consistent with our hypothesis, this analysis revealed that RPs regulate the expression of 493 genes

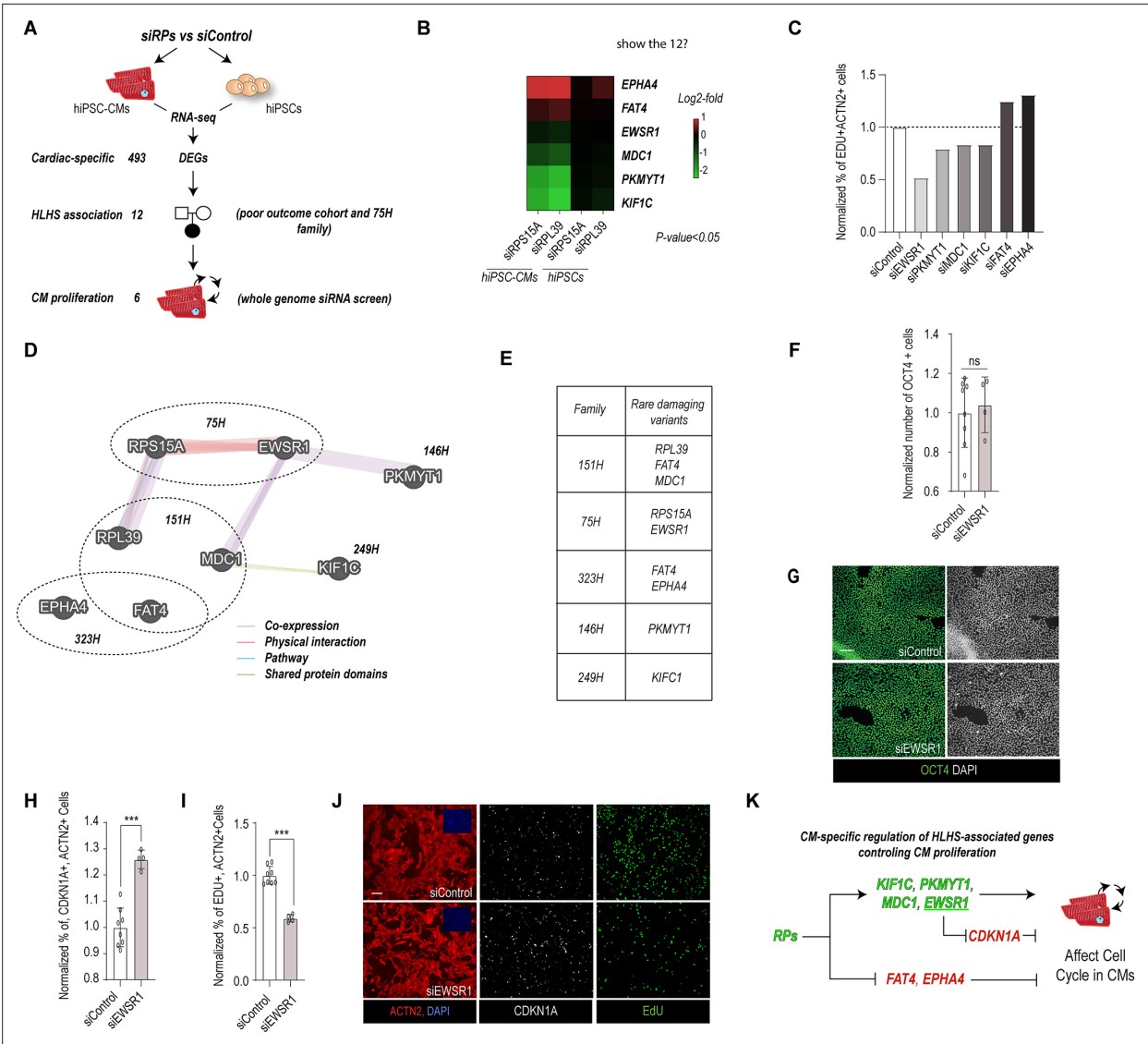

**Figure 7.** RP-dependent cardiac-specific regulation of cell proliferation. (**A**) Schematic illustrating approach to identify novel RP-dependent and cardiac-specific hypoplastic left heart syndrome (HLHS)-associated gene network controlling cardiomyocyte (CM) proliferation. (**B**) Heatmap showing differential expression (hiPSC-CMs vs hiPSCs) of genes regulating CM proliferation. (**C**) Histogram showing effect of cardiac-specific and HLHS-associated genes on CM proliferation. (**D**) Visualization of RP-dependent and cardiac-specific HLHS-associated gene network (GeneMania). HLHS families. (**E**) Table of HLHS families harboring rare and damaging variant in RP-dependent and cardiac-specific genes. (**F**) Histogram showing lack of effect of siEWSR1 on OCT4+ hiPSCs. (**G**) Representative images of OCT4+ cells in siControl and siEWSR1. (**H**) Histogram showing that siEWSR1 increases the % of CDKN1A+ CMs as compared to siControl. (**I**) Histogram showing that siEWSR1 concomitantly decreases the % of EDU+ CMs as compared to siControl. (**J**) Representative images showing immunostaining for CDKN1A (white), EDU (green), and ACTN2 (red) in siEWSR1 and siControl conditions. (**K**) Pathway reconstruction of cardiac and RP-dependent regulation of CM proliferation by HLHS-associated genes. Chi-square test, *p < 0.05, **p < 0.005, ***p < 0.001.

in a CM-specific manner (*Supplementary files 5–8*), thereby suggesting the existence of a cell type-specific (cardiac) transcriptional response to RP loss of function.

Next, to evaluate whether HLHS-associated genes were part of this lineage-specific transcriptional response, we selected those differentially expressed genes harboring rare and predicted-damaging variants in probands from the poor-outcome cohort and 75H family (*Supplementary files 2 and 4*). Remarkably, this prioritization strategy identified 12 genes potentially associated with HLHS (*Supplementary file 9*), 6 of which were found to also regulate cell cycle activity in hPSC-CMs (*Figure 7B, C*). Thus, collectively, this approach led us to identify an RP-dependent and cardiac-specific HLHS-associated gene network that supports CM proliferation *via* the upregulation of *EWSR1*, *MDC1*, *PKMYT1*, and *KIF1C*, and downregulation of *FAT4* and *EPHA4* (*Figure 7D, E*).

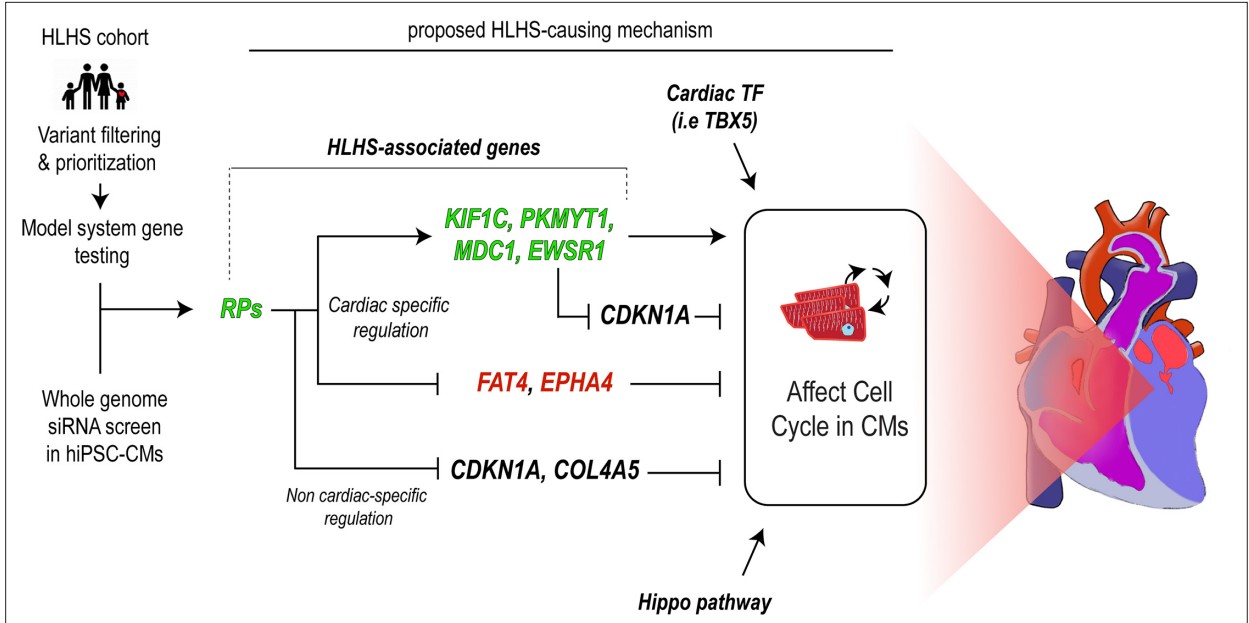

**Figure 8.** Model for RP-dependent involvement in hypoplastic left heart syndrome (HLHS)-associated phenotypes. Schematic showing that combined prioritization and unbiased screening led to the identification of a novel HLHS-associated gene network regulating cardiomyocyte (CM) proliferation as potential disease-causing mechanism.

The online version of this article includes the following figure supplement(s) for figure 8:

**Figure supplement 1.** RpS15Aa KD induces nucleolar stress in third instar larval hearts.

To further explore if RP-regulated HLHS-associated genes control proliferation in a cell type-specific manner, we asked if *EWSR1* KD, which elicited the strongest proliferation phenotype in hPSC-CMs, could regulate proliferation in non-cardiac cell types such as hPSCs. In contrast to hPSC-CMs, siEWSR1 did not affect the number of OCT4+ cells (*Figure 7F, G*), which suggests that EWSR1 is able to regulate proliferation CM-specifically. EWSR1 is an RNA/DNA-binding protein that regulates transcription and RNA splicing and plays a major role as oncogenic driver when fused to ETS transcription factors (*Flucke et al., 2021*). In the next step, we tested if EWSR1 regulates CM proliferation by modulating the p53 pathway. Consistent with this hypothesis, immunostaining for p53 downstream target, CDKN1A, revealed that EWSR1 KD both increased the percentage of CDKN1A-positive CMs and decreased EdU incorporation as compared to siControl (*Figure 7H–J*). Consistent with these observations, *RpS15Aa* deficiency fly line placed in trans to loss of function allele for *cabeza* (*caz*), the ortholog of *EWSR1*, caused prolongation of systolic interval lengths (*Figure 6—figure supplement 1F*) as compared to controls, suggesting a genetic interaction between these two genes. Collectively, these results support the conclusion that RPs can modulate proliferation in a cell type-specific manner, by controlling the expression of downstream effectors (i.e. *EWSR1*), with a cell type-specific ability to modulate the p53 pathway activity (*Figures 7k and 8*).

## Discussion

### Implication of RPs as genetic modulators in HLHS pathogenesis

Recent work, including our own, strongly suggests that defective cardiac differentiation and impaired CM proliferation contribute to HLHS-associated heart defects (*Liu et al., 2017*; *Gaber et al., 2013*; *Theis et al., 2020*). In this study, we find that RPs are critical for maintaining CM proliferation, as well as cardiac structure and function in fish and fly heart models. We show a potential role for RP function in cardiac development and HLHS pathogenesis as: (1) RPs are central regulators of CM proliferation; (2) RP variants are enriched in a cohort of poor-outcome HLHS patients; (3) a rare predicted-damaging promoter variant in *RPS15A* was associated with HLHS in familial CHD (75H); (4) KD of RPs in vivo results in cardiac deficiencies in hPSC-CMs, *Drosophila*, and zebrafish; and (5) induces TP53 signaling

and blocks proliferation that can be alleviated by inhibition of TP53 in the vertebrate models. Interestingly, the transcriptomic response of a previously reported HLHS proband (*Theis et al., 2020*) paralleled these findings and raised the possibility of an underlying ribosomopathy condition in that HLHS family. While universally important for cellular growth, an increasing number of variants in RP genes have been linked to CHD in humans, indicating tissue-specific differences in the penetrance of RP gene mutations. Most notably, ~30% of patients with Diamond-Blackfan anemia (*Vlachos et al., 2018*), a ribosomopathy characterized by hypo-proliferative, proapoptotic erythropoiesis, also have CHD. In agreement with another study (*Ikeda et al., 2017*), *rps15a* KD in fish caused a noticeable reduction in circulating red blood cells.

In addition to RPS15A, *RpL13* was recently identified as a potential candidate gene involved in CHD from a screen for de novo copy number variations in 167 patients with CHD (*Schroeder et al., 2019*). Furthermore, the Pediatric Cardiac Genomics Consortium (PCGC) exome dataset of 2871 patients with CHD identified several rare, predicted-damaging de novo variants in RP genes, 2 of which were found in patients with HLHS (*Jin et al., 2017*). Our finding that RPs genetically interact with core cTFs provides a potential mechanism for the cardiac specificity of RP phenotypes. In this context, we hypothesize that RP gene variants might play a role as genetic modifiers (or sensitizers) in the oligogenic pathogenesis of HLHS.

## Cardiac-specific and RP-dependent regulation of proliferation in the context of HLHS

Defective proliferation in LV CMs is a phenotypic hallmark of HLHS (*Liu et al., 2017*; *Gaber et al., 2013*), thereby implying the existence of HLHS-associated mechanisms that regulate proliferation in a CM-specific manner. In this study, we identified RPs as novel candidate genetic regulators of hypoplastic phenotypes observed in HLHS, as (1) rare, predicted-damaging variants affecting RP genes are enriched in HLHS probands as compared to healthy controls, (2) KD of most RPs impairs proliferation in hiPSC-CMs (59/80 tested) and heart development in flies (6/6 tested), and (3) systemic loss of *rps15A* function causes heart-specific hypoplastic phenotypes in zebrafish. However, in this context, the evaluation of RPs function in multiple cell types revealed that they are generally required for proliferation (hPSCs, hPSC-CMs, and dermal fibroblasts), thus indicating that RP loss of function triggers a non-cell type-specific cell cycle block.

Thus, an appealing mechanism by which RP function could be invoked in a spatially restricted manner would involve the regulation of RP expression by cell type-specific TFs. Consistent with this model, all probands harboring RP variants contain mutations affecting their promoter regions (*Table 1*). Also, several of these mutations are predicted to disrupt the binding site of previously identified HLHS-associated TF, ETS1 (*Miao et al., 2020*; *Ye et al., 2010*). Consistent with these observations, a recent study from the Bruneau lab (*Kathiriya et al., 2021*) shows that 38 RP genes are downregulated in TBX5-null hiPSC-CMs and ChIP analysis (*Ang et al., 2016*) reveals that a third of these dysregulated RPs (14/38 genes) are bound by TBX5 at their core promoter regions. Thus, we speculate that cell type-specific TFs controlling RP expression represent a promising gene class enabling the modulation of RP-dependent proliferation in a tissue-specific manner.

A second mechanism by which RPs might control proliferation and differentiation in a CM-specific manner is based on the hypothesis that CMs may be molecularly more sensitive to RP loss of function than other cell types. Consistent with this model, RP deficiency predisposes flies, fish, and hPSC-CM for interactions with core cardiogenic TFs, including TBX5/*Doc*, GATA4/*pnr*, and NKX2–5/*tin*, the latter being the most cardiac restricted. (*Figure 6*). Moreover, our comparative transcriptomics analysis (*Figure 4*) revealed that RPs regulate gene expression in a CM-specific manner. Among these RP-regulated genes, *EWSR1*, *MDC1*, *PKMYT1*, *KIF1C*, *FAT4*, and *EPHA4*, were found to be mutated in HLHS families and regulate CM proliferation (*Figure 7*). Remarkably, EWSR1 KD did not affect proliferation in hPSCs, while it strongly reduced EdU incorporation in hPSC-CMs, thus suggesting that EWSR1 regulates proliferation in a CM-specific manner. Interestingly, in flies, *RpS15Aa* genetically interacts with *cabeza* (*caz*, the fly ortholog of *EWSR1*). Consistent with our observations, a recent study has found that overexpression of a fusion Ewsr1-Fli1 protein in mice specifically induces dilated cardiomyopathy by promoting apoptosis in cardiac myocytes (*Tanaka et al., 2015*). Similarly, Fat4, a negative regulator of the Hippo pathway, which is upregulated upon RP KD, was found to specifically regulate heart size via the regulation of CM size and proliferation (*Ragni et al., 2017*). Thus, collectively, we

propose that RPs represent a central regulatory hub for cell proliferation and differentiation during embryonic development, which can be modulated in a cell type-specific manner by tissue-restricted modulators of RP expression and/or activity.

## An RP–MDM2–p53/YAP surveillance network in heart development

Defects in ribosome biogenesis caused, for example, by reduction in RP gene copy numbers induce a stress response program that activates TP53 (p53) and a cascade of cellular events resulting in apoptosis, inhibition of the cell cycle, or DNA damage response (reviewed in e.g. *Danilova and Gazda, 2015*), commonly referred to as nucleolar stress response (*Boulon et al., 2010*) with activation of TP53 being a key hallmark (*Yang et al., 2018*). A key mechanism involves the release of RPs to the nucleoplasm, where several RPs bind and inhibit MDM2, which again results in p53 stabilization and pathway activation (*Dai et al., 2004*; *Zhang and Lu, 2009*). A recent study showed that RP haploinsufficiency in the developing mouse limb bud also activates a common TP53 cascade but results in TP53-dependent tissue-specific changes of the translatome, which might confer the specificity often observed in ribosomopathy (*Tiu et al., 2021*). The TP53-MDM2 feedback loop is the central molecular node in response to a wide variety of stress signals, including in the human diseased adult heart (*Men et al., 2021*; *Mak et al., 2017*). As mentioned above, the transcriptomic response of the HLHS family 5H (*Theis et al., 2020*) also showed activation of the TP53 pathway potentially due to an underlying ribosomopathy. Since low RP levels induce a proliferation blockade that can be overridden by p53-p21/CDKN1A KD, we hypothesize that RP levels may act as signaling components sensing cellular fitness and with a functional outcome either being p53 activation or non-activation. Such an RP–MDM2–p53 surveillance network was previously proposed to be important in response to nutrient availability changes and inhibition of oncogenic activity (*Deisenroth and Zhang, 2011*; *Liu et al., 2016*). Thus, the inhibition of the RP–MDM2–p53 axis might be a therapeutic avenue to consider for functional intervention, although further studies are needed to identify the upstream mechanisms leading to the *p53* pathway activation observed in HLHS proband cells.

Heart-specific KD of the *RpS15Aa* in *Drosophila* causes constriction of larval hearts and atrophy in adult hearts, due to heart loss during metamorphosis, which could not be rescued by p53 reduction, unlike in hPSC-CMs and zebrafish. This might be due to a different, context-dependent role of *p53* in flies previously reported to cause a dwarfing, *Minute*-like phenotype (*Cui and DiMario, 2007*). We hypothesized that the loss of RPs in flies, in conjunction with reduced protein synthesis, might cause nucleolar stress, triggering a cell-intrinsic signaling cascade that prevents the heart from further differentiating and growing. To determine whether cardiac KD of *RpS15Aa* causes nucleolar stress in the *Drosophila* heart, we stained larval hearts for Fibrillarin, a marker for nucleoli and nucleolar integrity. We found that *RpS15Aa* KD causes expansion of nucleolar Fibrillarin staining in CM, which is a hallmark of nucleolar stress (*Figure 8—figure supplement 1A–C*). As a control, we also performed cardiac KD of *Nopp140*, which is known to cause nucleolar stress upon loss-of-function. We found a similar expansion of Fibrillarin staining in larval CM nuclei (*Figure 8—figure supplement 1C, D*). This suggests that *RpS15Aa* KD indeed causes nucleolar stress in the *Drosophila* heart, which likely contributes to the dramatic heart loss in adults.

The role of the Hippo pathway to regulate cell growth/organ size, proliferation, and survival, and its importance in cardiac biology is conserved between humans and fly (*Barron and Kagey, 2014*; *Del Re, 2014*; *Yu et al., 2015*; *Wang et al., 2018*). The Hippo–YAP pathway is a cell-intrinsic pathway that regulates CM proliferation and thus heart size during development, as demonstrated by the ability of activated Yap to induce postnatal CM regeneration (*Xiao et al., 2016*). Interestingly, we find here that overexpression of the *Drosophila* ortholog of YAP/TAZ, Yorkie, rescues *RpS15Aa* KD-induced heart loss, dependent on its downstream factor *scalloped* (the *Drosophila* ortholog of *TEAD1/2/3/4*) (*Figure 5*). Of note, YAP1 is not only an important regulator of CM proliferation in the embryo but also promotes CM survival and growth in the postnatal heart (*von Gise et al., 2012*; *Del Re et al., 2013*), which is in line with our findings. Interestingly, a proband among the PCGC HLHS patients was transheterozygous for mutations in *RPL15* and *TEAD4* (*Jin et al., 2017*), making this an interesting disease candidate pair.

## Rapid gene discovery and prioritization approach for complex genetic diseases

Uncovering the genetic basis of polygenic and heterogeneous diseases, such as HLHS, remains a significant challenge. This difficulty arises from the lack of experimental approaches capable of rapidly determining the role and contribution of genetic variants, as well as their epistatic relationships, in generating disease-associated phenotypes. The current state-of-the-art approach involves using CRISPR-mediated editing or correction of gene variants (*Gifford et al., 2019*), which is effective for analyzing single-family pedigrees. However, this method is impractical for prioritizing the large number of candidate genes identified through cohort-wide analyses. Consequently, progress in gene discovery for HLHS has been limited in recent years (*Yagi et al., 2018*; *Birla et al., 2022*). In this study, we aimed to address these limitations by employing a HT and unbiased exploration of genes regulating a conserved HLHS-relevant phenotype (*Gaber et al., 2013*; *Miao et al., 2020*; *Theis et al., 2020*; *Xu et al., 2022*), such as reduced CM proliferation, in hiPSCs. Notably, results integration from a whole-genome functional screen in hiPSC-derived CMs (*Figure 1*, RPs are top hits from the screen), genomic data from an HLHS parent–proband cohort (*Figure 2*, RPs represent the most enriched gene class containing rare damaging variants), hiPSC-CM phenotyping from a high-value HLHS family (75H), and functional validation in two independent in vivo model systems (*Figure 3*), identified RPs as a novel class of HLHS-associated genes. While our study highlights the potential of this approach for gene prioritization, additional research is needed to directly demonstrate the functional consequence of the identified genetic variants, to verify an association between RP encoding genes and HLHS in other patient cohorts with and without poor outcome, and determine if RP variants have a broader role in CHD susceptibility. In conclusion, we propose that the approach outlined in this study provides a novel framework for rapidly prioritizing candidate genes and systematically testing them, individually or in combination, using a CRISPR/Cas9 genome-editing strategy in mouse embryos (*Cunningham et al., 2017*). Applied in the context of complex genetic diseases, this framework has the potential to yield deeper insights into their underlying mechanisms.

## Materials and methods

### Key resources table

| Reagent type (species) or resource | Designation | Source or reference | Identifiers | Additional information |
|---|---|---|---|---|
| Genetic reagent (*D. melanogaster*) | *Hand*[4.2]-Gal4 | Bodmer lab | PMID:16467358 | |
| Genetic reagent (*D. melanogaster*) | R94C02::tdTomato | N. Jan lab | FBtp0137272 | |
| Genetic reagent (*D. melanogaster*) | UAS-RpS15Aa[RNAi] | Vienna *Drosophila* Resource Center (VDRC) | FBgn0010198 | v19198 |
| Genetic reagent (*D. melanogaster*) | Df(RpS15Aa) | Bloomington *Drosophila* Stock Center (BDSC) | FBab0047266 | 39614 |
| Genetic reagent (*D. melanogaster*) | UAS-RpL26[RNAi] | Vienna *Drosophila* Resource Center (VDRC) | FBgn0036825 | v40402 v100280 |
| Genetic reagent (*D. melanogaster*) | UAS-RpL36A[RNAi] | Vienna *Drosophila* Resource Center (VDRC) | FBgn0031980 | v108391 |
| Genetic reagent (*D. melanogaster*) | UAS-RpS15[RNAi] | Vienna *Drosophila* Resource Center (VDRC) | FBgn0034138 | v35415 v104439 |
| Genetic reagent (*D. melanogaster*) | UAS-RpL39[RNAi] | Vienna *Drosophila* Resource Center (VDRC) | FBgn0023170 | v23578 v108821 |
| Genetic reagent (*D. melanogaster*) | UAS-RpL3[RNAi] | Vienna *Drosophila* Resource Center (VDRC) | FBgn0020910 | v109820 |
| Genetic reagent (*D. melanogaster*) | UAS-yorkie | Pan | | |
| Genetic reagent (*D. melanogaster*) | UAS-Myc[RNAi] | Bloomington *Drosophila* Stock Center (BDSC) | FBgn0262656 | 25784 |
| Genetic reagent (*D. melanogaster*) | UAS-sd[RNAi] | Vienna *Drosophila* Resource Center (VDRC) | FBgn0003345 | v101497 |

*Continued on next page*

*Continued*

| Reagent type (species) or resource | Designation | Source or reference | Identifiers | Additional information |
|---|---|---|---|---|
| Genetic reagent (*D. melanogaster*) | *Df(3L)DocA* | Reim | Fbab0037663 | |
| Genetic reagent (*D. melanogaster*) | tin$^{EC40}$ | Bloomington *Drosophila* Stock Center (BDSC) | Fbal0032861 | 78560 |
| Genetic reagent (*D. melanogaster*) | tin$^{346}$ | Bloomington *Drosophila* Stock Center (BDSC) | Fbal0035787 | 92964 |
| Genetic reagent (*D. melanogaster*) | pnr$^{VX6}$ | Bloomington *Drosophila* Stock Center (BDSC) | Fbal0032468 | 6334 |
| Genetic reagent (*D. melanogaster*) | Df(pnr) | Bloomington *Drosophila* Stock Center (BDSC) | Fbab0038315 | 7982 |
| Strain, strain background (*Danio rerio*) | Oregon AB wild-type | Ocorr lab, SBP | | A commonly used wild-type strain |
| Strain, strain background (*Danio rerio*) | *Tg(myl7:EGFP)*$^{twu277}$ | Tsai Lab, National Taiwan University | PMID:12950077 | A transgenic line of zebrafish labeled with heart-specific EGFP fluorescence |
| Strain, strain background (*Danio rerio*) | *Tg(myl7:H2A-mCherry)*$^{sd12}$ | Yelon Lab, University of California, San Diego | PMID:24075907 | A transgenic line of zebrafish specifically expressing mCherry in cardiomyocyte nuclei |
| Antibody | Mouse monoclonal anti-ACTN1 | Sigma | A7811 | 1:800 |
| Antibody | Mouse monoclonal anti-POU5F1 (OCT4) | Sigma | P0082 | 1:500 |
| Antibody | Donkey polyclonal anti-mouse Alexa Fluor 568 | Invitrogen | A10037 | 1:500 |
| Antibody | Chicken polyclonal anti-GFP | Aves Labs | GFP-1020 | 1:300 |
| Antibody | Rabbit polyclonal anti-mCherry | Rockland | 600-401P16S | 1:200 |
| Antibody | Donkey polyclonal anti-chicken AlexaFluor 488 | Jackson ImmunoResearch | 703-545-155 | 1:200 |
| Antibody | Donkey polyclonal anti-rabbit AlexaFluor 568 | Invitrogen | A10042 | 1:200 |
| Other | DAPI (iPSC) 500 mg/ml | Sigma | D9542 | Nuclear stain 1:1000 |
| Antibody | Mouse anti-Mhc (*Drosophila*) | DSHB | 3E8-3D3 | 1:50 |
| Antibody | Anti-mouse-Alexa Fluor 488 | Jackson Labs | 115-545-003 | 1:500 |
| Antibody | Alexa Fluor 647 phalloidin | Invitrogen | A22287 | 1:500 |
| Other | DAPI (Zebrafish) 500 mg/ml | Invitrogen | D1306 | Nuclear stain 1:200 |
| Sequence-based reagent | RPS15A siRNA | Entrez Gene ID: 6210 | Dharmacon | On-Target plus, individual sequence |
| Sequence-based reagent | TP53 siRNA | Entrez Gene ID: 7157 | Dharmacon | On-Target plus, Individual Sequence |
| Sequence-based reagent | CDKN1A siRNA | Entrez Gene ID: 1026 | Dharmacon | On-Target plus, SmartPool |
| Sequence-based reagent | TP53 | 371502118c1 | IDT Integrated DNA Technologies, Coralville, IA | Expression level |

*Continued on next page*

*Continued*

| Reagent type (species) or resource | Designation | Source or reference | Identifiers | Additional information |
|---|---|---|---|---|
| Sequence-based reagent | CDKN1A | 310832423c1 | IDT Integrated DNA Technologies, Coralville, IA | Expression level |
| Sequence-based reagent | CCNB1 | 356582356c1 | IDT Integrated DNA technologies, Coralville, IA | Expression level |
| Sequence-based reagent | CCNB2 | 332205979c1 | IDT Integrated DNA technologies, Coralville, IA | Expression level |
| Sequence-based reagent | CDK1 | 281427275c1 | IDT Integrated DNA technologies, Coralville, IA | Expression level |
| Sequence-based reagent | MCM2 | 33356546c1 | IDT Integrated DNA technologies, Coralville, IA | Expression level |
| Sequence-based reagent | RPS15A | 71772358c2 | IDT Integrated DNA Technologies, Coralville, IA | Expression level |
| Sequence-based reagent | GAPDH | Hs.PT.45.8326 | IDT Integrated DNA Technologies, Coralville, IA | Expression level |
| Commercial assay or kit | EdU | Click-it Plus EdU Imaging Kit | Life Technologies | |
| Software, algorithm | Prism v7 and v8 | SBP license | GraphPad Software | |

## Study subjects

Written informed consent was obtained for HLHS probands and family members under a research protocol approved by the Mayo Clinic Institutional Review Board. The studies described in this manuscript were conducted according to the guidelines of the Declaration of Helsinki and approved by the Institutional Review Board of the Mayo Clinic (HLHS protocol 11-000114 approved 10 March 2011).

Cardiac anatomy was assessed by echocardiography. Candidate genes were identified and prioritized by WGS of genomic DNA and RNA-sequencing of patient-specific iPSC and CMs. Methods for genomic analyses, RNA-sequencing, iPSC technology, bioinformatics, and statistics are described in the Online Appendix.

## WGS and bioinformatic strategies

Our methods for family phenotyping, WGS, variant filtering, and candidate gene prioritization in the rare HLHS-CHD family and 25 HLHS proband–parent trios have been previously described (*Theis et al., 2015a*; *Theis et al., 2015b*; *Bodmer, 1993*). In brief, WGS of DNA isolated from blood or cheek swabs was performed on a HiSeq 4000 platform at the Mayo Clinic Medical Genome Facility. BAM files underwent primary and secondary analyses using an established workflow with standard data quality metrics, and reads were aligned to the human hg38 reference genome. Variant call format files with single nucleotide variants and small insertion–deletion (indel) calls were analyzed using Ingenuity Variant Analysis software and knowledge database (QIAGEN). To retain only high-confidence data, variants were required to have a base call quality of at least 20 and to pass Variant Quality Score Recalibration. Functionally annotated coding and regulatory variants underwent primary filtering and prioritization by an iterative approach with variants required to have an MAF <0.01 across all races in gnomAD v2.1 (*Karczewski et al., 2020*). Variants predicted to impact protein structure were retained and included missense, frameshift, stop-gain, stop-loss, canonical splice, and non-canonical variants predicted to disrupt splicing based upon MaxEntScan (*Grodecká et al., 2014*). Regulatory variants were defined as those that (a) impact a microRNA or microRNA-binding site, (b) reside in an enhancer binding site annotated in the VISTA database (http://enhancer.lbl.gov/), or (c) disrupt a predicted promoter or transcription factor-binding site informed by the position-weighted matrices available in JASPAR (http://jaspar.cgb.ki.se/). Genes harboring a rare functional variant were further required to

have upper quartile cardiac expression during mouse embryonic (e14.5) or human fetal heart development ranked percentiles from RNA-seq experiments available in ENCODE: ENCSR047LLJ (120-day male); ENCSR863BUL (91-day female); ENCSR000AEZ (28- and 19-week female).

For the HLHS-CHD family, a secondary segregation filter was applied to model autosomal dominant inheritance with incomplete penetrance, which required sharing of prioritized rare variants between affected family members. For the poor-outcome HLHS proband–parent trios, a secondary Mendelian filter was applied to identify major-effect driver variants that arose de novo or fit recessive modes of inheritance (i.e., homozygosity, hemizygosity, compound heterozygosity).

To determine whether certain gene networks were over-represented, two online bioinformatics tools were used. First, STRING (*Szklarczyk et al., 2019*) was used to investigate experimental and predicted protein–protein and genetic interactions, and clustering of RP genes was demonstrated when the highest stringency filter was applied. In addition, PANTHER (*Mi et al., 2019*) was employed to identify potential enrichment of genes by ontology classification.

## Generation of hPSC-VCMs

Id1 overexpressing hPSCs (derived from *Burridge et al., 2015*) were dissociated with 0.5 mM EDTA (Thermo Fisher Scientific) in PBS without $CaCl_2$ and $MgCl_2$ (Corning) for 7 min at room temperature (RT). hPSCs were resuspended in mTeSR-1 media (StemCell Technologies) supplemented with 2 µM Thiazovivin (StemCell Technologies) and plated in a Matrigel-coated 12-well plate at a density of $3 \times 10^5$ cells per well. After 24 hr after passage, cells were fed daily with mTeSR-1 media (without Thiazovivin) for 3–5 days until they reached ≥90% confluence to begin differentiation. hPSC-VCMs were differentiated as previously described (*Burridge et al., 2015*). At day 0, cells were treated with 6 µM CHIR99021 (Selleck Chemicals) in S12 media (*Pei et al., 2017*) for 48 hr. At day 2, cells were treated with 2 µM Wnt-C59 (Selleck Chemicals), an inhibitor of WNT pathway, in S12 media. Forty-eight hours later (at day 4), S12 media is fully changed. At day 5, cells were dissociated with TrypLE Express (Gibco) for 2 min and blocked with RPMI (Gibco) supplemented with 10% FBS (Omega Scientific). Cells were resuspended in S12 media supplemented with 4 mg/l Recombinant Human Insulin (Gibco) (S12+ media) and 2 µM Thiazovivin and plated onto a Matrigel-coated 12-well plate at a density of $9 \times 10^5$ cells per well. S12+ media was changed at day 8 and replaced at day 10 with RPMI (Gibco) media supplemented with 213 µg/µl L-ascorbic acid (Sigma), 500 mg/l BSA-FV (Gibco), 0.5 mM L-carnitine (Sigma), and 8 g/l AlbuMAX Lipid-Rich BSA (Gibco; CM media). Typically, hPSC-ACMs start to beat around day 10. At day 15, cells were purified with lactate media RPMI without glucose, 213 µg/µl L-ascorbic acid, 500 mg/l BSA-FV, and 8 mM sodium-DL-Lactate (Sigma), for 4 days. At day 19, media was replaced with CM media.

## siRNA transfection, proliferation assay, and immunostaining in hPSC-VCMs

At day 25 of differentiation, hPSC-VCMs were dissociated with TrypLE Select 10X (Gibco), 10 min and neutralized with RPMI supplemented with 10% FBS. Cells were resuspended in RPMI with 2% KOSR (Gibco) and 2% B27 50X with vitamin A (Life Technologies) supplemented with 2 µM Thiazovivin and plated at a density of 5000 cells/well in a Matrigel-coated 384-well plate. hPSC-VCMs were transfected with siRNA (Dharmacon: ON-TARGETplus, custom RNAi cherry-pick libraries 0.1 nmol). For the whole-genome screening, siRNAs directed to 18,000 human genes were purchased from the Genomic Center Facility at SBP at a final concentration of 25 nM using lipofectamine RNAiMax (Thermo Fisher) and opti-MEM (Gibco). Forty-eight hours post-transfection, cells were labeled with 10 µM EdU (Thermo Fisher). After 24 hr of EdU incubation, cells were fixed with 4% paraformaldehyde for 30 min and blocked in blocking buffer (10% Horse Serum, 10% Gelatin, and 0.5% Triton X-100) for 20 min. EdU was detected according to protocol and cells were stained with cardiac-specific marker ACTN2 (A7811, Sigma, dilution 1/800), secondary antibody Alexa Fluor 568 (Invitrogen, 1/500) and DAPI (1/1000) in Blocking Buffer. Cells were imaged with ImageXpress Micro XLS microscope (Molecular Devices) and custom algorithms were used to quantify % of EdU+ ACTN1+ hPSC-VCMs and the number of ACTN1+ cells. To quantify TP53 levels in hPSC-VCMs, cells were stained with Phospho-p53 (700439, Thermo Fisher, 1/500), counterstained with ACTN1 (anti-mouse A7811, Sigma, or anti-rat ab50599, Abcam), and imaged and quantified with ImageXpress microscope. Whole-genome

screening was performed in one replicate. All the other siRNA experiments were performed in quadruplicates.

## RNA-seq and data analysis

hPSCs and hPSC-derived CMs were transfected with 25 nM final concentrations of siRNA against RPS15A, RPL39, and with scrambled control siRNAs as above. Two days after siRNA transfection, RPs KD were verified and confirmed by proliferation assay (see above), and cells were pelleted and resuspended in 500 µl TRIzol reagent followed by total RNA extraction. Library preparation and sequencing of the samples was done at La Jolla Institute of Immunology (La Jolla, CA). FASTQ files were processed using nf-core/rnaseq (version 21.03.0.edge; *Ewels et al., 2020*). Differential gene expression was determined using R/DESeq2 (*Love et al., 2014*) and GO term enrichment was done using gprofiler2 (*Raudvere et al., 2019*). Analysis scripts can be downloaded at https://github.com/gvogler/elife-2025-nielsen-et-al (copy archived at *Vogler, 2025*). Raw reads and counts tables are available at GEO accession number GSE207658.

## Proliferation assay in parent/proband iPSCs

hPSCs were dissociated as described previously and plated in a Matrigel-coated 384-well plate at a density of 3000 cells per well. Cells were transfected with siRNA at a final concentration of 25 nM using lipofectamine RNAiMax and opti-MEM. After 3 days, cells were labeled with 10 µM EdU for 1 hr. Cells were fixed with 4% paraformaldehyde for 30 min and blocked in blocking buffer for 20 min. EdU was detected according to the protocol and stained with stem cell-specific marker OCT4 (P0082, Sigma, 1/500), secondary antibody Alexa Fluor 568 and DAPI in Blocking Buffer. Cells were imaged with ImageXpress microscope, and % of EdU+ cells and number of OCT4+ cells were quantified.

## Quantitative real-time PCR (RT-qPCR) in parent/proband iPSCs and hPSC-CMs

Total RNA was extracted using TRIzol and chloroform. 1 µg of RNA was converted to cDNA using QuantiTect Reverse Transcription kit (QIAGEN). qRT-PCR was performed using SYBR green (Bio-Rad). Human primer sequences for qRT-PCR were obtained from Harvard Primer Bank: TP53 (Primer Bank ID: 371502118c1), CDKN1A (Primer Bank ID: 310832423c1), CCNB1 (Primer Bank ID: 356582356c1), CCNB2 (Primer Bank ID: 332205979c1), CDK1 (Primer Bank ID: 281427275c1), MCM2 (Primer Bank ID: 33356546c1), RPS15A (Primer Bank ID: 71772358c2). All values were normalized to *GAPDH* (*Primer Bank ID: 378404907c1*). At least three independent biological replicates were performed for each experiment.

## Statistical analysis

To determine any statistical significance between experimental and control groups in hPSC-VCMs and iPSCs from parent/proband trio, we calculated two-sided p values with Student's *t*-test using GraphPad Prism 8.1.2 software.

## *Drosophila* heart function studies

*Drosophila* orthologs were determined using the DIOPT database (*Hu et al., 2011*), and fly stocks were obtained from the Vienna *Drosophila* Resource Center (VDRC) stock center or Bloomington *Drosophila* Stock Center (BDSC) as indicated in the key source table. For in vivo functional heart analysis, we developed a high-throughput method based on genetically modified flies with CM-specific RFP fluorescence. The reporter line by itself or combined with the heart-specific *Hand*[4.2]-Gal4 driver including all cardioblasts/CMs, pericardial cells, and wing hearts throughout development starting at post-mitotic, mid-embryonic stages through adulthood (*Han et al., 2006*; *Tögel et al., 2013*; *Hallier et al., 2015*) was crossed to UAS-lines or mutant flies. For interaction studies, a fly line harboring the fluorescent reporter, *Hand*[4.2]-Gal4 driver, and a deficiency for RpS15Aa was generated. Adult progeny flies were immobilized and exposed to fluorescence light to record 5 s high-frame-rate movies of the beating heart. Movies were analyzed by fully automated quantification of contractility and rhythmicity parameters of the heart (*Vogler, 2021*). Semi-intact adult fly hearts were filmed and analyzed according to standard protocol (*Fink et al., 2009*; *Cammarato et al., 2015*).

## Immunostainings of the fly heart

The immunostaining of fly adult hearts was performed as described previously (*Alayari et al., 2009*). In short, adult flies were dissected in a semi-intact fashion to expose the heart according to protocol (*Fink et al., 2009*; *Ocorr et al., 2014*). Myofibrils were relaxed using 10 mM EGTA followed by fixation in 4% formaldehyde for 15 min. Hearts were washed with PBS + Triton (PBT; 0.03% Triton X-100) and stained using Alexa 647 phalloidin (Life Technologies) 2 hr at RT. For Mhc labeling, hearts were stained using anti-Mhc antibody (1:50, incubation overnight at 4°C) and after three washes with PBT, secondary antibody was applied for 2 hr at RT. Hearts were washed with PBT three times and PBS one time and mounted using ProLong Gold mountant with DAPI (Life Technologies). Heart preparations were imaged with the Imager.Z1 with an Apotome (Carl Zeiss), Hamamatsu Orca Flash 4.0, and ZEN imaging software (Carl Zeiss).

## Zebrafish husbandry

All zebrafish experiments were performed in accordance with protocols approved by IACUC, AUF 22-073. Zebrafish were maintained under standard laboratory conditions at 28.5°C. In addition to Oregon AB wild-type, the following transgenic lines were used: *Tg(myl7:EGFP)*[twu277] (*Huang et al., 2003*) and *Tg(myl7:H2A-mCherry)*[sd12] (*Schumacher et al., 2013*).

In zebrafish, gene expression was manipulated using standard microinjection of MO antisense oligonucleotides (*Wang et al., 2018*). In addition, we performed targeted mutagenesis using CRISPR/Cas9 genome editing (*Talbot and Amacher, 2014*; *Gagnon et al., 2014*; *Irion et al., 2014*), to create insertion/deletion (INDEL) mutations in relevant ribosomal genes ($F_0$). Subsequently, zebrafish were raised to 72 hr post-fertilization (hpf), immobilized in low melt agarose, and the hearts were filmed and analyzed according to standard protocol (*Fink et al., 2009*).

## MO sequence information

All MOs were purchased from Gene Tools, LLC, synthesized at 300 nmol, except zebrafish p53 oligos at 100 nmol.

> Zebrafish *rpl13* 5'-UTR MO: TTGTTCACTCCGTCCTTAGCGGAAA
> Zebrafish *rps15a* 5'-UTR MO: CGCACCATGATGCCAGTTCTGCAAT
> Zebrafish *rpl39* 5'-UTR MO: GGATCGCAATCCGTTCACCACTATG
> Zebrafish p53 MO: GCGCCATTGCTTTGCAAGAATTG
> Zebrafish Control MO: 5'-CCT CTT ACC TCA GTT ACA ATT TAT A-3'.

## Zebrafish SOHA (semi-automated optical heartbeat analysis)

Larval zebrafish (72 hpf) were immobilized in a small amount of low melt agarose (1.5%) and submerged in conditioned water. Beating hearts were imaged with direct immersion optics and a digital high-speed camera (up to 200 frame/s, Hamamatsu Orca Flash) to record 30 s movies; images were captured using HC Image (Hamamatsu Corp). Cardiac function was analyzed from these high-speed movies using semi-automatic optical heartbeat analysis software (*Fink et al., 2009*; *Ocorr et al., 2009*), which for zebrafish quantifies heart period (R–R interval), cardiac rhythmicity, as well as chamber size and fractional area change. All hearts were imaged at RT (20–21°C). Statistical analyses were performed using Prism software (GraphPad). Significance was determined using two-tailed, unpaired Student *t*-test or one-way ANOVA and Dunnett's multiple comparisons post hoc test as appropriate.

## Zebrafish CM cell counts and cardiac immunofluorescent imaging

To count CMs, we used the expression of H2AmCherry in the nuclei (*Tg(myl7:H2A-mCherry)*) (*Schumacher et al., 2013*) to qualify as an individual cell, performed the 'Spot' function in Imaris to distinguish individual cells in reconstructions of confocal z-stacks. (*Zeng and Yelon, 2014*; *Pradhan et al., 2017*). To compare datasets, we used Prism software (GraphPad) to perform Student's *t*-test with two-tail distribution. Graphs display mean and standard deviation for each dataset.

Whole-mount immunofluorescence was performed as previously described (*Theis et al., 2020*; *Zeng and Yelon, 2014*; *Pradhan et al., 2017*; *Alexander et al., 1998*) (see also key resources table). Confocal imaging was performed on an LSM 710 confocal microscope (Zeiss, Germany) with a 40x

water objective. Exported z-stacks were processed with Imaris software (Bitplane), Zeiss Zen, and Adobe Creative Suite software (Photoshop and Illustrator 2020). All confocal images shown are projection views of partial reconstructions from multiple z-stack slices, except where noted that images are views of a single slice.

For proliferation assays, embryos at 24, 48, and 72 hpf were fixed in 4% paraformaldehyde overnight at 4°C, washed in PBS with 0.1% Tween-20 (PBST). Samples were blocked in 5% normal goat serum in PBST prior to incubation with primary antibodies. Proliferating cells were labeled with anti-phospho-histone H3 (Sigma-Aldrich-H0412 PH3; 1:500) and total nuclei were counterstained with DAPI. Embryos were mounted in ProLong Gold for imaging. Whole-mount hearts were imaged on a Zeiss Apotome microscope at ×10 magnification. Z-stacks were acquired and reconstructed using Fiji/ImageJ. Regions corresponding to atrium, ventricle, atrioventricular canal, and outflow tract were manually segmented. For some experiments, CM nuclei were identified based on Tg(myl7:H2A-mCherry) expression, with DAPI used to quantify total cell counts. For proliferation experiments, DAPI was used to identify cardiac cells, and proliferating cells were identified by PH3 staining. Proliferative indices were calculated as PH3+/DAPI+ ratios from z-stacks analyzed in Fiji/ImageJ using the Cell Counter plugin. Cell counts were performed on at least 10 embryos per condition per timepoint.

### Zebrafish CRISPR/Cas9 experiments

Detailed steps were previously described (*Hoshijima et al., 2019*) and we followed IDT manufacture instruction for complexes preparation. crRNA:tracrRNA Duplex Preparation:Target-specific Alt-R crRNA (sequence information see below) and common Alt-R tracrRNA were synthesized by IDT and each RNA was dissolved in duplex buffer (IDT) as 100 µM stock solution. Stock solutions were stored at –20°C. To prepare the crRNA:tracrRNA duplex, equal volumes of 100 µM Alt-R crRNA and 100 µM Alt-R tracrRNA stock solutions were mixed together and annealed by heating followed by gradual cooling to RT by manufacture instruction: 95°C, 5 min on PCR machine; cool to 25°C; cool to 4°C rapidly on ice. The 50 µM crRNA:tracrRNA duplex stock solution was stored at –20°C. Preparation of crRNA:tracrRNA:Cas9 RNP complexes: Cas9 protein (Alt-R S.p. Cas9 nuclease, v.3, IDT) was adjusted to 25 µM stock solution in 20 mM HEPES-NaOH (pH 7.5), 350 mM KCl, 20% glycerol, dispensed as 8 µl aliquots, and stored at –80°C. 25 µM crRNA:tracrRNA duplex was produced by mixing equal volumes of 50 µM crRNA:tracrRNA duplex stock and duplex buffer (IDT). We used 5 µM RNP complex. To generate 5 µM crRNA:tracrRNA:Cas9 RNP complexes: 1 µl 25 µM crRNA:tracrRNA duplex was mixed with 1 µl 25 µM Cas9 stock, 2 µl $H_2O$, and 1 µl 0.25% phenol red solution. Prior to microinjection, the RNP complex solution was incubated at 37°C for 5 min and then placed at RT. Approximately one nanoliter of 5 µM RNP complex was injected into the cytoplasm of one-cell stage embryos to generate $F_0$ larva.

### IDT crRNA sequence information

All crRNAs were purchased from Integrated DNA Technologies, Inc, synthesized at 2 nmol with standard desalting condition.

> Dr.Cas9.RPS15A.1.AC: /AltR1/rUrU rGrUrU rGrUrC rArArU rCrUrC rArCrA rGrGrG rGrUrU rUrUrA rGrArG rCrUrA rUrGrC rU/AltR2/
> Dr.Cas9.RPS15A.1.AB: /AltR1/rGrC rGrUrA rCrUrA rUrGrA rCrUrU rUrArG rArGrC rGrUrU rUrUrA rGrArG rCrUrA rUrGrC rU/AltR2/
> Dr.Cas9.RPS17.1.AC: /AlTR1/rUrGrArCrUrUrCrCrArCrArUrUrArArCrArArGrCrGrUrUrUrUrArGrArGrCrUrArUrGrCrU/AlTR2/
> Dr.Cas9.RPS28.1.AA: /AlTR1/rCrUrGrGrGrArArGrArArCrUrGrGrCrUrCrCrCrArGrUrUrUrUrArGrArGrCrUrArUrGrCrU/AlTR2/

### Acknowledgements

This work was supported by a grant from the Wanek Foundation at Mayo Clinic in Rochester, MN, to JLT, TJN, TMO, RB, and ARC. We gratefully acknowledge the patients and families who participated in this study. We thank Marco Tamayo for excellent technical assistance. This work was supported by National Institutes of Health (R01 HL054732 to RB.; R01 HL153645, R01 HL148827, R01 HL149992,

R01 AG071464 to ARC); by California Institute for Regenerative Medicine (DISC2-10110 to ARC); by the American Heart Association (AHA Predoctoral Fellowship 18PRE33960593 to KB).

## Additional information

### Funding

| Funder | Grant reference number | Author |
|---|---|---|
| National Heart Lung and Blood Institute | HL153645 | Alexandre R Colas |
| National Heart Lung and Blood Institute | HL148827 | Alexandre R Colas |
| National Heart Lung and Blood Institute | HL054732 | Rolf Bodmer |
| National Institutes of Health | R01HL149992 | Alexandre R Colas |
| National Institutes of Health | R01AG071464 | Alexandre R Colas |
| California Institute for Regenerative Medicine | DISC2-10110 | Alexandre R Colas |
| American Heart Association | AHA Predoctoral Fellowship 18PRE33960593 | Katja Birker |

The funders had no role in study design, data collection, and interpretation, or the decision to submit the work for publication.

### Author contributions

Tanja Nielsen, Anaïs Kervadec, Maria A Missinato, James Marchant, Michaela Romero, Katya Marchetti, Aashna Lamba, Xin-Xin I Zeng, Marie Berenguer, Stanley M Walls, Analyne Schroeder, Katja Birker, Investigation; Jeanne L Theis, Formal analysis; Greg Duester, Paul Grossfeld, Methodology; Timothy J Nelson, Conceptualization; Timothy M Olson, Conceptualization, Supervision, Writing – original draft, Writing – review and editing; Karen Ocorr, Georg Vogler, Conceptualization, Supervision, Investigation, Writing – original draft, Writing – review and editing; Rolf Bodmer, Alexandre R Colas, Conceptualization, Supervision, Funding acquisition, Writing – original draft, Writing – review and editing

### Author ORCIDs

Jeanne L Theis ⓘ https://orcid.org/0000-0002-4494-8683
Katya Marchetti ⓘ https://orcid.org/0009-0000-9635-3246
Marie Berenguer ⓘ https://orcid.org/0000-0002-3629-9505
Timothy J Nelson ⓘ https://orcid.org/0000-0002-3862-7023
Timothy M Olson ⓘ https://orcid.org/0000-0003-2716-9423
Karen Ocorr ⓘ https://orcid.org/0000-0003-2593-0119
Rolf Bodmer ⓘ https://orcid.org/0000-0001-9087-1210
Georg Vogler ⓘ https://orcid.org/0000-0002-8303-3531
Alexandre R Colas ⓘ https://orcid.org/0000-0001-8489-0570

### Ethics

Human studies described in this manuscript were conducted according to the guidelines of the Declaration of Helsinki and approved by the Institutional Review Board of the Mayo Clinic (HLHS protocol 11-000114 approved 10 March 2011). Informed consent was obtained from all participants, and all work complied with the Declaration of Helsinki and relevant privacy regulations, including HIPAA. De-identified samples were used whenever possible to ensure confidentiality.

All experiments were conducted in accordance with institutional, national, and international ethical guidelines. Zebrafish studies were approved by the Institutional Animal Care and Use Committee (IACUC) of SBP under AUF 22-073 and were performed in accordance with the U.S. Public Health Service Policy on Humane Care and Use of Laboratory Animals and the NIH Guide for the Care and

Use of Laboratory Animals. Animals were housed in accredited facilities and all procedures, including anesthesia and euthanasia, were carried out to minimize pain, distress, and the number of animals used.

Reviewer #1 (Public review): https://doi.org/10.7554/eLife.106231.3.sa1
Reviewer #2 (Public review): https://doi.org/10.7554/eLife.106231.3.sa2
Author response https://doi.org/10.7554/eLife.106231.3.sa3

# Additional files

## Supplementary files
Supplementary file 1. Whole-genome siRNA screen results table.

Supplementary file 2. 292 filtered variants from poor-outcome HLHS probands. MOI – mode of inheritance.

Supplementary file 3. Subset from all 292 genes that cause CM proliferation defects and/or fly heart defects.

Supplementary file 4. Segregating variants of 75H proband.

Supplementary file 5. Differential gene expression lists for siRPL39 and siRPS15A in hPSCs and CMs.

Supplementary file 6. Differential gene expression lists for siRPL39 and siRPS15A in hPSCs and CMs.

Supplementary file 7. Differential gene expression lists for siRPL39 and siRPS15A in hPSCs and CMs.

Supplementary file 8. Differential gene expression lists for siRPL39 and siRPS15A in hPSCs and CMs.

Supplementary file 9. Twelve genes prioritized after RNA-seq and whole-genome sequencing (WGS) analysis.

MDAR checklist

## Data availability
For RNA-sequencing, raw reads and counts tables are available at GEO accession number GSE207658.

The following dataset was generated:

| Author(s) | Year | Dataset title | Dataset URL | Database and Identifier |
|---|---|---|---|---|
| Vogler G | 2022 | siRPS15A-3 | https://www.ncbi.nlm.nih.gov/geo/query/acc.cgi?acc=GSM6304454 | NCBI Gene Expression Omnibus, GSM6304454 |

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
