## [Editor Report · eLife Assessment]

This **important** study applies an innovative multi-model strategy to implicate the ribosomal protein (RP) encoding genes as candidates causing Hypoplastic Left Heart Syndrome. The evidence from the screen in stem cell-derived cardiomyocytes and whole genome sequencing of human patients, followed by functional analyses of RP genes in fly and fish models, is **convincing** and supports the authors' claims. This work and methodology applied would be of broad interest to medical biologists working on congenital heart diseases.

---

## [Referee Report · Reviewer #1 (Public review)]

Nielsen et al have identified a new disease mechanism underlying hypoplastic left heart syndrome due to variants in ribosomal protein genes that lead to impaired cardiomyocyte proliferation. This detailed study starts with an elegant screen in stem cell derived cardiomyocytes and whole genome sequencing of human patients and extends to careful functional analysis of RP gene variants in fly and fish models. Striking phenotypic rescue is seen by modulating known regulators of proliferation including the p53 and Hippo pathways. Additional experiments suggest that cell type specificity of the variants in these ubiquitously expressed genes may result from genetic interactions with cardiac transcription factors. This work positions RPs as important regulators of cardiomyocyte proliferation and differentiation involved in the etiology of HLHS, and point to potential downstream mechanisms.

The revised manuscript has been extended, facilitating interpretation and reinforcing the authors' conclusions.

---

## [Referee Report · Reviewer #2 (Public review)]

Tanja Nielsen et al. presents a novel strategy for identification of candidate genes in Congenital Heart Disease (CHD). Their methodology, which is based on comprehensive experiments across cell models, drosophila and zebrafish models, represents an innovative, refreshing and very useful set of tools for identification of disease genes, in a field which are struggling with exactly this problem.

The authors have applied their methodology to investigate the pathomechanisms of Hypoplastic Left Heart Syndrome (HLHS) - a severe and rare subphenotype in the large spectrum of CHD malformations. Their data convincingly implicates ribosomal proteins (RPs) in growth and proliferation defects of cardiomyocytes, a mechanism which is suspected to be associated with HLHS.

By whole genome sequencing analysis of a small cohort of trios (25 HLHS patients and their parents) the authors investigated a possible association between RP encoding genes and HLHS.

Although the possible association between defective RPs and HLHS needs to be verified, the results suggest a novel disease mechanism in HLHS, which is a potentially substantial advance in our understanding of HLHS and CHD. The conclusions of the paper are based on solid experimental evidence from appropriate high- to medium-throughput models, while additional genetic results from an independent patient cohort is needed to verify an association between RP encoding genes and HLHS in patients.

---

## [Author Response]

The following is the authors’ response to the original reviews.

**Reviewer #1 (Public review):**
Nielsen et al have identified a new disease mechanism underlying hypoplastic left heart syndrome due to variants in ribosomal protein genes that lead to impaired cardiomyocyte proliferation. This detailed study starts with an elegant screen in stemcell-derived cardiomyocytes and whole genome sequencing of human patients and extends to careful functional analysis of RP gene variants in fly and fish models. Striking phenotypic rescue is seen by modulating known regulators of proliferation, including the p53 and Hippo pathways. Additional experiments suggest that the cell type specificity of the variants in these ubiquitously expressed genes may result from genetic interactions with cardiac transcription factors. This work positions RPs as important regulators of cardiomyocyte proliferation and differentiation involved in the etiology of HLHS, although the downstream mechanisms are unclear.

We thank Reviewer 1 for the thoughtful assessment of our manuscript. Our point-bypoint responses to the recommendations are provided (Reviewer 1, “Recommendations for the authors”).

**Reviewer #2 (Public review):**
Tanja Nielsen et al. present a novel strategy for the identification of candidate genes in Congenital Heart Disease (CHD). Their methodology, which is based on comprehensive experiments across cell models, Drosophila and zebrafish models, represents an innovative, refreshing and very useful set of tools for the identification of disease genes, in a field which are struggling with exactly this problem. The authors have applied their methodology to investigate the pathomechanisms of Hypoplastic Left Heart Syndrome (HLHS) - a severe and rare subphenotype in the large spectrum of CHD malformations. Their data convincingly implicates ribosomal proteins (RPs) in growth and proliferation defects of cardiomyocytes, a mechanism which is suspected to be associated with HLHS.By whole genome sequencing analysis of a small cohort of trios (25 HLHS patients and their parents), the authors investigated a possible association between RP encoding genes and HLHS. Although the possible association between defective RPs and HLHS needs to be verified, the results suggest a novel disease mechanism in HLHS, which is a potentially substantial advance in our understanding of HLHS and CHD. The conclusions of the paper are based on solid experimental evidence from appropriate high- to medium-throughput models, while additional genetic results from an independent patient cohort are needed to verify an association between RP encoding genes and HLHS in patients.

We thank Reviewer 2 for the thoughtful assessment of our manuscript. Our point-by-point responses to the recommendations are provided (Reviewer 2, “Recommendations for the authors”).

**Reviewer #1 (Recommendations for the authors):**
(1) Despite an interesting surveillance model, the disease-causing mechanisms directly downstream of the RP variants remain unclear. Can the authors provide any evidence for abnormal ribosomes or defects in translation in cells harboring such variants? The possibility that reduced translation of cardiac transcription factors such as TBX5 and NKX2-5 may contribute to the functional interactions observed should be considered. How do the authors consider that the RP variants are affecting transcript levels as observed in the study?

Our model implies that cell cycle arrest does not require abnormal ribosomes or translational defects but instead relies on the sensing of RP levels or mutations as a fitness-sensing mechanism that activates TP53/CDKN1A-dependent arrest. Supporting this framework, we observed no significant changes in TBX5 or NKX2-5 expression (data not shown), but rather an upregulation of CDKN1A levels upon RP KD.

(2) The authors suggest that a nucleolar stress program is activated in cells harboring RP gene variants. Can they provide additional evidence for this beyond p53 activation?

We added additional data to support nucleolar stress (Suppl. Fig. 6) and text (lines 52635):

To determine whether cardiac KD of *RpS15Aa* causes nucleolar stress in the *Drosophila* heart, we stained larval hearts for Fibrillarin, a marker for nucleoli and nucleolar integrity. We found that *RpS15Aa* KD causes expansion of nucleolar Fibrillarin staining in cardiomyocyte, which is a hallmark of nucleolar stress (Suppl. Fig. 6A-C). As a control, we also performed cardiac KD of *Nopp140*, which is known to cause nucleolar stress upon loss-of-function. We found a similar expansion of Fibrillarin staining in larval cardiomyocyte nuclei (Suppl. Fig. 6C,D). This suggests that *RpS15Aa* KD indeed causes nucleolar stress in the *Drosophila* heart, that likely contributes to the dramatic heart loss in adults.

Other recommendations:
*(3) Concerning the cell type specificity, in the proliferation screen, were similar effects seen on the actinin negative as actinin positive EdU+ cells? It would be helpful to refer to the fibroblast result shown in Supplementary Figure 1C in the results section.*

As suggested by reviewer #1, we have added a reference to Supplementary Fig. 1C, D and noted that RP knockdown exerts a non–CM-specific effect on proliferation.

(4) The authors refer to HLHS patients with atrial septal defects and reduced right ventricular ejection fraction. Please clarify the specificity of the new findings to HLHS versus other forms of CHD, as implied in several places in the manuscript, including the abstract.

This study focused on a cohort of 25 HLHS proband-parent trios selected for poor clinical outcome, including restrictive atrial septal defect and reduced right ventricular ejection fraction. We have revised the following sentence in response to the Reviewer’s comment (lines 567-571): “While our study highlights the potential of this approach for gene prioritization, additional research is needed to directly demonstrate the functional consequence of the identified genetic variants, verify an association between RP encoding genes and HLHS in other patient cohorts with and without poor outcome, and determine if RP variants have a broader role in CHD susceptibility.

(5) The multi-model approach taken by the authors is clearly a good system for characterizing disease-causing variants. Did the authors score for cardiomyocyte proliferation or the time of phenotypic onset in the zebrafish model?

We used an antibody against phosphohistone 3 to identify proliferating cells and DAPI to identify all cardiac cells in control injected, rps15a morphants, and rps15a crispants. We found that cell numbers and proliferating cells were significantly reduced at 24 and 48 hpf. By 72 hpf cardiac cell proliferation is greatly diminished even in controls, where proliferation typically declines.

Reduced ventricular cardiomyocyte numbers could potentially result from impaired addition of LTPB3-expressing progenitors. In experiments where altered cardiac rhythm is observed, please comment on the possible links to proliferation.

Heart function data showed that heart period (R-R interval) was unaffected in morphants and crispants at 72 hpf where we also observed significant reductions in cell numbers. This suggests that the bradycardia observed in the rps15a + nkx2.5 or tbx5a double KD (Sup. Fig. 5D & E) was not due to the reduction in cell numbers alone.

Finally, the use of the mouse to model HLHS in potential follow-up studies should be discussed.

We have added a mouse model comment to the discussion (lines 571-74): “In conclusion, we propose that the approach outlined in this study provides a novel framework for rapidly prioritizing candidate genes and systematically testing them, individually or in combination, using a CRISPR/Cas9 genome-editing strategy in mouse embryos (PMID: 28794185)”.

(6) When the authors scored proliferation in cells from the proband in family 75H, did they validate that RPS15A expression is reduced, consistent with a regulatory region defect?

Good point. We examined RPS15A expression in these cells and found no significant reduction in gene expression in day 25 cardiomyocytes (data not shown). One possible explanation is that this variant may regulate RPS15A expression in a stage-specific manner during differentiation or under additional stress conditions.

(7) Minor point. Typo on line 494: comma should be placed after KD, not before.

Thank you, this has now been corrected (new line 490)

**Reviewer #2 (Recommendations for the authors):**
(1) The authors are invited to revise the part of the manuscript that describes the genetic analysis and provide a more balanced discussion of the WGS data, with a conclusion that aligns with the strength of the human genetic data.

We disagree with reviewer #2’s assessment. The goal of our study is not to apply a classical genetic approach to establish variant pathogenicity, but rather to employ a multidisciplinary framework to prioritize candidate genes and variants and to examine their roles in heart development using model systems. In this context, genetic analysis serves primarily as a filtering tool rather than as a means of definitively establishing causality.

(2) The genetic analysis of patients does not appear to provide strong evidence for an association between RP gene variants and HLHS. More information regarding methodology and the identified variants is needed.

HLHS is widely recognized as an oligogenic and heterogeneous genetic disease in which traditional genetic analyses have consistently failed to prioritize any specific gene class as reviewer#2 is pointing out. Therefore, relying solely on genetic analysis is unlikely to yield strong evidence for association with a given gene class. This limitation provides the rationale for our multidisciplinary gene prioritization strategy, which leverages model systems to interrogate candidate gene function. Ultimately, definitive validation of this approach will require studies in relevant in vivo models to establish causality within the context of a four-chambered heart (see also Discussion).

In Table S2, it would be appropriate to provide information on sequence, MAF, and CADD. Please note the source of MAF% (GnomAD version?, which population?).

As summarized in Figure 2A, the 292 genes from the families with the 25 proband with poor outcome displayed in Supplemental Table 2 fulfilled a comprehensive candidate gene prioritization algorithm based on the variant, gene, inheritance, and enrichment, which required all of the following: (1) variants identified by whole genome sequencing with minor allele frequency <1%; (2) missense, loss-of-function, canonical splice, or promoter variants; (3) upper quartile fetal heart expression; and (4) De novo or recessive inheritance. Unbiased network analysis of these 292 genes, which are displayed in Supplemental Table 2 for completeness, identified statistically significant enrichment of ribosomal proteins. The details about MAF, CADD score, and sequence highlighted by the Reviewer are provided for the RP genes in Table 1, which are central to the focus and findings of the manuscript.

It would also be helpful for the reader if genome coordinates (e.g., 16-11851493-G-A for RSL1D1 p.A7V) were provided for each variant in both Table 1 and S2.

Genome coordinates have been added to Table 1.

(3) The dataset from the hPSC-CM screen could be of high value for the community. It would be appropriate if the complete dataset were made available in a usable format.

The dataset from the hPSC-CM screen has been added to the manuscript as Supp Table 1

(4) The "rare predicted-damaging promoter variant in RPS15A" (c.-95G>A) does not appear so rare. Considering the MAF of 0,00662, the frequency of heterozygous carriers of this variant is 1 out of 76 individuals in the general population. Thus, considering the frequency of HLHS in the population (2-3 out of 10,000) and the small size of family 75H, the data do not appear to indicate any association between this particular variant and HLHS. The variants in Table 1 also appear to have relatively mild effects on the gene product, judging from the MAF and CADD scores. The authors are invited to discuss why they find these variants disease-causing in HLHS.

Our study design is based on the widely held premise that HLHS is an oligogenic disorder. Our multi-model systems platform centered on comprehensive filtering of coding and regulatory variants identified by whole genome sequencing of HLHS probands to identify candidate genes associated with susceptibility to this rare developmental phenotype. 75H proved to be a high-value family for generating a relatively short list of candidate genes for left-sided CHD. Given the rarity of both left-sided CHD and the *RPS15A* variant identified in the HLHS proband and his 5th degree relative, with a frequency consistent with a risk allele for an oligogenic disorder, we made the reasonable assumption that this was a *bona fide* genotype-phenotype association rather than a chance occurrence. Moreover, incomplete penetrance and variable expression is consistent with a genetically complex basis of disease whereby the shared variant is risk-conferring and acts in conjunction with additional genetic, epigenetic, and/or environmental factors that lead to a left-sided CHD phenotype. In sum, we do not claim these variants are definitively disease causing, but rather potentially contributing risk factors.

(5) Information is lacking on how clustering of RP genes was demonstrated using STRING (with P-values that support the conclusions). What is meant by "when the highest stringency filter was applied"? Does this refer to the STRING interaction score or something else? The authors could also explain which genes were used to search STRING (e.g., all 292 candidate genes) and provide information on the STRING interaction score used in the analysis, the number of nodes and edges in the network.

To determine whether certain gene networks were over-represented, two online bioinformatics tools were used. First, genes were inputted into STRING (Author response image 2) to investigate experimental and predicted protein-protein and genetic interactions. Clustering of ribosomal protein genes was demonstrated when applying the highest stringency filter. Next, genes were analyzed for potential enrichment of genes by ontology classification using PANTHER .Applying Fisher’s exact test and false discovery rate corrections, ribosomal proteins were the most enriched class when compared to the reference proteome, including data annotated by molecular function (4.84-fold, p=0.02), protein class (6.45-fold, p=0.00001), and cellular component (9.50fold, p=0.001). A majority of the identified RP candidate genes harbored variants that fit a recessive inheritance disease model.

**Author response image 2. sa3fig2:**